



# The impact of El Niño–Southern Oscillation on the total column ozone over the Tibetan Plateau

Yang Li[1,2], Wuhu Feng[2,3], Xin Zhou[1,2], Yajuan Li[4,5], Martyn P. Chipperfield[2,6]

[1]School of Atmospheric Sciences, Chengdu University of Information Technology, Chengdu, China
[2]School of Earth and Environment, University of Leeds, Leeds, UK
[3]National Centre for Atmospheric Science, University of Leeds, Leeds, UK
[4]School of Electronic Engineering, Nanjing Xiaozhuang University, Nanjing, China
[5]Key Laboratory of Middle Atmosphere and Global Environment Observation, Institute of Atmospheric Physics, Chinese Academy of Sciences, Beijing, China
[6]National Centre for Earth Observation, University of Leeds, Leeds, UK

*Correspondence to*: Yang Li (Y.Li10@leeds.ac.uk) and Wuhu Feng (W.Feng@ncas.ac.uk)

**Abstract.** The Tibetan Plateau (TP, approximately 27.5–37.5°N, 75.5–105.5°E) is the highest and largest plateau on Earth with a mean elevation of over 4 km. This special geography causes strong surface solar ultraviolet radiation (UV), with

potential risks to human and ecosystem health, and which is controlled by the local total column ozone (TCO). The El Niño–Southern Oscillation (ENSO), the dominant mode of interannual variability on Earth, is characterized by warming of central and eastern equatorial Pacific Ocean sea surface temperature anomalies (SSTA) for the El Niño phase and cooling for the La Niña phase. Although some studies have suggested that there exists a positive correlation between ENSO and the TP TCO, the mechanism underlying this effect of ENSO is not fully understood.

Here we use the Copernicus Climate Change Service (C3S) merged satellite dataset and the TOMCAT 3–dimensional (3D) offline chemical transport model forced by ERA5 meteorological reanalyses from the European Centre for Medium–Range Weather Forecasts (ECMWF) over the period 1979–2021 to investigate the influence of ENSO on the TCO over the TP. The correlation coefficient of the time series of monthly TCO anomalies over the TP between the TOMCAT output and C3S dataset is ~0.92 with statistical significance above 99%. In particular, the correlation coefficients (December–May) are above 0.95,

indicating that the TCO variability in TOMCAT is very consistent with the merged satellite observations. Based on the lead correlation between the ENSO index (Niño 3.4 index from the National Oceanic and Atmospheric Administration) and TP TCO time series, we find that ENSO has a significant impact on the TCO, especially from the December of its mature phase to the May of its decaying phase (December–May). That is, the El Niño (La Niña) events induce positive (negative) TCO anomalies over the TP. Through studying the ozone profile from the Stratospheric Water and OzOne Satellite Homogenized

(SWOOSH) dataset and TOMCAT results over their overlapping period (1984–2021), we attribute the positive (negative) TCO anomalies mainly to the increased (decreased) ozone at the 200–70 hPa levels caused by the downward (upward) shift of ozone profile associated with the lower (higher) tropopause height during the El Niño (La Niña) events. Our results suggest that the El Niño events lead to the descending upper–level geopotential height and hence cause the decreasing air column



thickness, which in turn induces the cooling tropospheric temperature over the TP. This cooling associated with El Niño events causes a decrease of the tropopause height, which contributes to the downward shift of the ozone profile and hence leads to the increase in TCO. The La Niña events affect TP TCO during December–May in a manner resembling the El Niño events, except with anomalies of opposite sign.

This work provides a systematic understanding of the influence of ENSO on ozone over the TP. Since climate models project an increase in the frequency of strong El Niño and La Niña events under greenhouse–gas–forced warming, we can expect more ozone variation associated with ENSO, with important implications for 21st–century ozone recovery, surface solar UV, and ecosystems over the TP.

## 1 Introduction

The Tibetan Plateau (TP) broadly extends over the latitude–longitude domain of 27.5–37.5°N, 75.5–105.5°E (Li et al., 2020), with a size of about one–quarter of the Chinese territory (Wu et al., 2007). It is the highest and largest plateau on Earth, commonly termed the "Roof of the World" (Royden et al., 2008), and plays a key role in dominating the atmospheric circulation in Asia through dynamical and thermal forcing (e.g. Yeh, 1950; Flohn, 1957; Yanai et al., 1992; Wu et al., 2012). Due to its high elevation, low air density, and high atmospheric transparency, the TP experiences strong surface solar ultraviolet (UV) radiation, whose excess can cause harmful influences on the local biota (Liu et al., 2016). Atmospheric ozone absorbs most incoming solar UV radiation thereby protecting living organisms at the surface (Staehelin et al., 2001). Therefore, there is a strong interest in better understanding the processes controlling ozone over the TP.

Zhou et al. (1995) found that there is a significant total column ozone low (TOL) centered over the TP during summer from Total Ozone Mapping Spectrometer (TOMS) satellite measurements. Over the past decades, several studies have focused on total column ozone (TCO) and its associated TOL over the TP during summer (e.g. Zou, 1996; Cong et al., 2002; Liu et al., 2003; Ye and Xu, 2003; Zheng et al., 2004; Bian et al., 2006; Tian et al., 2008; Tobo et al., 2008; Guo et al., 2012). These studies have argued that TOL is caused by mass exchange between troposphere and stratosphere (Cong et al., 2002), convective activity (Liu et al., 2003), high topography and thermal forcing of the TP (Kiss et al., 2007; Ye and Xu, 2003), anticyclone in the upper troposphere (Zheng et al., 2004), variations in tropopause height (Tian et al., 2008), Asian summer monsoon (Bian et al., 2011), South Asian high (Guo et al., 2012), and quasi–biennial oscillation (QBO, Chang et al., 2022). Meanwhile, Zhang et al. (2014) pointed out that the TOL over the TP during winter and spring has deepened over the recent decades, which is associated with the upward movement of the tropopause due to the rapid and significant warming over the TP.

Apart from the QBO (e.g. Fusco and Salby, 1999; Kiss et al., 2007), the interannual variability of the TCO changes over the TP is closely linked with El Niño–Southern Oscillation (ENSO). ENSO represents a periodic fluctuation between warmer (El Niño) and colder (La Niña) sea surface temperature (SST) conditions over the central and eastern tropical Pacific (e.g. Wallace et al., 1998; McPhaden et al., 2006; Li et al., 2017; Zhang et al., 2019). As a prominent interannually varying natural



phenomenon, ENSO has pronounced climate impacts around the globe (e.g. van Loon and Madden, 1981; Ropelewski and
      Halpert, 1987; Trenberth et al., 1998). Most studies on ENSO impacts on ozone interannual variability have focused on the
      tropical stratosphere (e.g., Shiotani, 1992; Hasebe, 1993; Randel et al., 2009; Oman et al., 2011; Xie et al., 2014; Olsen et al.,
      2016), and on the polar region (e.g. Cagnazzo et al., 2009; Domeisen et al., 2019), with these regions showing the significant
      interannual variability. ENSO teleconnection to the mid–latitude ozone changes, especially ozone changes over the TP, which
is the focus of this study, has been less discussed. Earlier studies by Zou et al. (2001) suggest the amplitude of ENSO signal
      in TCO over the TP is about 20 DU (their figure 3), but their results are based on very limited satellite measurements from
      1979 to 1992. Positive correlations between ENSO index and TCO during winter, spring, and summer are supported by recent
      studies using longer ozone observations (Zhou et al., 2013; Zhang et al., 2015a; Li et al., 2020; Chang et al., 2022). However,
      a systematic understanding of how ENSO influences TCO over TP is still lacking. Here we use a much wider range and longer
period of ozone measurements, along with a chemical transport model specific to the years of observation, to provide a first
      extensive examination of ENSO signal in TCO changes over the TP.

      After a brief description of the data, model, and methods in Section 2, Section 3 presents the findings of the work, focusing on
      addressing the three key questions: (1) How long can the wintertime ENSO signal persist in TCO changes over the TP? (2)
      Where is its largest signal in the vertically resolved ozone changes? (3) How does ENSO impact ozone changes in this region?
Finally, our summary and discussion are presented in Section 4. The overall goal of this study is to provide a more accurate
      estimation of ENSO impacts on ozone interannual changes over the TP. With this knowledge, we can have a better
      understanding on the mid–latitude ozone changes leading to a better estimation of the ozone recovery (from decreasing levels
      of stratospheric chlorine and bromine) in this region.

## 2 Data and methods

### 2.1 Observation-based datasets

      The merged satellite TCO observation is from the Copernicus Climate Change Service (C3S) product during the satellite era
      (1979–2021). This data is available at https://cds.climate.copernicus.eu/cdsapp#!/dataset/satellite-ozone-v1?tab=overview
      (last access: May 2023), and a more detailed description and validation results on TCO data are available at
      https://datastore.copernicus-climate.eu/documents/satellite-ozone/C3S2_312a_Lot2_PUGS_O3_latest.pdf (last access: May
2023). This data is created by combining total ozone data from 15 satellite sensors, including the Global Ozone Monitoring
      Experiment (GOME, 1995–2011), Scanning Imaging Absorption Spectrometer for Atmospheric Chartography
      (SCIAMACHY, 2002–2012), Ozone Monitoring Instrument (OMI, 2004–present), GOME-2A/B (2007–present), Backscatter
      Ultraviolet Radiometer (BUV-Nimbus4, 1970–1980), Total Ozone Mapping Spectrometer (TOMS-EP, 1996–2006), series of
      Solar Backscatter Ultraviolet Radiometers (SBUV, 1985–present), and Ozone Mapping and Profiler Suite (OMPS, 2012–
present). The horizontal resolution of C3S data is 0.5° latitude × 0.5° longitude. The long-term stability of the TCO product is
      within the 1% per decade level, and its systematic error is below 2%.



The Stratospheric Water and OzOne Satellite Homogenized (SWOOSH) version 2.6 dataset (Davis et al., 2016) is used to study the ozone profiles. It is a merged record of stratospheric ozone and water vapour measurements and consists of data from the Stratospheric Aerosol and Gas Experiment (SAGE-II/III), Upper Atmospheric Research Satellite Halogen Occultation Experiment (UARS HALOE), UARS Microwave Limb Sounder (MLS), and Aura MLS instruments. The measurements of SWOOSH are homogenized by applying corrections that are calculated from data taken during time periods of instrument overlap (Davis et al., 2016). The merged SWOOSH record with 5° latitude × 20° longitude horizontal resolution spans from 1984 to the present, and has 12 levels per decade in pressure ranging from 316 to 1 hPa (31 pressure levels). The SWOOSH data and a more detailed description are available at https://csl.noaa.gov/groups/csl8/swoosh/ (last access: May 2023).

We use the monthly European Centre for Medium–Range Weather Forecasts (ECMWF) recent fifth generation reanalysis (ERA5) (Hersbach et al., 2020) to investigate the atmospheric circulation and temperature as well as tropopause. Here we choose the ERA5 product at a resolution of 1.0° latitude × 1.0° longitude and over an altitude range from 1000 to 0.01 hPa, which is available at https://cds.climate.copernicus.eu/cdsapp#!/dataset/reanalysis-era5-pressure-levels-monthly-means?tab=form (last access: May 2023). The monthly SST from the Hadley Centre Sea Ice and SST dataset version 1 (HadISST1) with a resolution of 1.0° latitude × 1.0° longitude (Rayner et al., 2003) is used. The HadISST1 data and a more detailed description are available at https://www.metoffice.gov.uk/hadobs/hadisst/ (last access: May 2023). The monthly Niño 3.4 index from the National Oceanic and Atmospheric Administration's (NOAA) Climate Prediction Center (CPC) is used in this study to represent ENSO, which is based on the monitoring of area averaged of SST anomalies (SSTA) in the central and eastern equatorial Pacific (5°S–5°N, 120°W–170°W). The Niño 3.4 index is available at https://psl.noaa.gov/gcos_wgsp/Timeseries/Nino34/ (last access: May 2023). The El Niño and La Niña events by seasons from NOAA's CPC is used to classify cold and warm episodes of ENSO, which is available at https://origin.cpc.ncep.noaa.gov/products/analysis_monitoring/ensostuff/ONI_v5.php (last access: May 2023). Except for SWOOSH, the overlapping period of all data is 1979–2021 and that the anomalies represent the deseasonalised anomalies with respect to the 1979–2021. As the SWOOSH spans from 1984 to the present, its anomalies are with respect to the 1984–2021.

## 2.2 TOMCAT model

Here we use the three–dimensional (3D) global offline chemical transport model (TOMCAT/SLIMCAT; hereafter TOMCAT) which is described in detail by Chipperfield (2006). The TOMCAT performs well in reproducing the observed ozone variations (e.g. Feng et al., 2005; Singleton et al., 2005; Rösevall et al., 2008; Kuttippurath et al., 2010; Chipperfield et al., 2017, 2018; Griffin et al., 2019; Bognar et al., 2021; Feng et al., 2021; Li et al., 2022). The TOMCAT model has a detailed stratospheric chemistry scheme (e.g. Feng et al., 2011; Chipperfield et al., 2018; Grooß et al., 2018; Weber et al., 2021), including the major ozone-depleting substances (ODSs) and greenhouse gases (Carpenter et al., 2018), aerosol effects from volcanic eruptions (e.g. Dhomse et al., 2015), and variations in solar forcing (e.g. Dhomse et al., 2013; 2016). The model was forced using winds and temperatures from meteorological reanalysis (usually ECMWF) to specify the atmospheric transport and temperatures and



calculates the abundances of chemical species in the troposphere and stratosphere. In this study, we used the latest ECMWF
ERA5 reanalysis product (Hersbach et al., 2020) to force the model. The TOMCAT simulations are performed at 2.8° latitude
× 2.8° longitude and have 32 levels from the surface to 60 km. The model was run for the period 1950–2022. Note that the
overlapping period with observations (i.e., 1979–2021) is used for analysis.

**2.3 Methods**

Several statistical methods have been applied in this study. For the tropopause height, we follow the same definition based on
the World Meteorological Organization (WMO, 1957), which is identified by the temperature lapse rate. Note that this
definition has been shown to be one of the best identifications (Maddox and Mullendore, 2018). Lead–lag correlation
coefficient and composite analysis are applied to investigate the impact of ENSO on ozone over the TP. The statistical
significance of the correlation between two auto–correlated time series is calculated using the two–tailed Student's $t$–test and
the effective number ($N^{\text{eff}}$) of degrees of freedom (Pyper and Peterman, 1998; Li et al., 2013), as given by the following
approximation:

$$\frac{1}{N^{\text{eff}}} \approx \frac{1}{N} + \frac{2}{N}\sum_{j=1}^{N}\frac{N-j}{N}\rho_{XX}(j)\rho_{YY}(j) \qquad (1)$$

where $N$ is the sample size, and $\rho_{XX}$ and $\rho_{YY}$ are the autocorrelations of two sampled time series, $X$ and $Y$, respectively, at time
lag $j$.

Following the hypsometric equation, the mean temperature of the atmospheric layer between the pressure $p_1$ and $p_2$ can be
written as follows (Wallace et al., 1996; Holton and Hakim, 2013; Sun et al., 2017; Li et al., 2022):

$$\langle T\rangle = \frac{g_0}{R}\left(\ln\frac{p_1}{p_2}\right)^{-1}\Delta Z,$$
$$\Delta Z = Z_2 - Z_1 \qquad (2)$$

where $\langle T\rangle$ is the mean temperature of the atmospheric layer, $\Delta Z$ is the thickness of layer, $Z_1$ and $Z_2$ are, respectively, the
geopotential heights at $p_1$ and $p_2$, $g_0 = 9.80665$ m s$^{-2}$ is the global average of gravity at mean sea level, and $R = 287$ J kg$^{-1}$ K$^{-1}$
is the gas constant for dry air. If we let $\langle T\rangle'$ and $\Delta Z'$ be the deviations or anomalies from their time average, then equation (2)
changes to the perturbation hypsometric equation as follows:

$$\langle T\rangle' = \frac{g_0}{R}\left(\ln\frac{p_1}{p_2}\right)^{-1}\Delta Z' \qquad (3)$$

The above equation (3) suggests that the anomalous mean temperature of the layer is proportional to the anomalous thickness
of the layer bounded by isobaric surfaces. Therefore, the mean temperature should decrease (increase) if the perturbed layer



thins (thickens). Following previous studies (e.g. Wallace et al., 1996; Sun et al., 2017; Li et al., 2022; Zhang et al., 2022), we
use this perturbation equation (3) to discuss the influences of atmospheric thickness on tropospheric temperature.

## 3 Results

In this section, we first examine whether the TOMCAT can reproduce the TCO variability over the TP by the comparison with the merged satellite TCO from C3S dataset. **Figure 1** shows the correlation coefficients between the time series of 3–month averaged TCO anomalies over the TP (27.5–37.5°N, 75.5–105.5°E) for the C3S dataset and TOMCAT over the period 1979–
2021. This averaged TP region is identical to Li et al. (2020). All correlation coefficients between the C3S dataset and TOMCAT in **Figure 1** are strong (above 0.65) and are statistically significant at the 99% level based on the two–tailed Student's $t$–test and the $N^{\mathrm{eff}}$ of degrees of freedom defined in equation (2). In particular, for DJF (December–January–February), JFM (January–February–March), FMA (February–March–April), and MAM (March–April–May), the correlation coefficients are higher (above 0.95, **Figure 1**), indicating that the TP TCO variability of TOMCAT is consistent with that of C3S dataset
from December to May. It is also noted that the correlation coefficient of monthly time series of TP TCO between C3S dataset and TOMCAT is about 0.92 and is statistically significant at the 99% level. These results indicate that TOMCAT reproduces well the observed TCO variability over the TP, especially from December to May. Therefore, it is reasonable to use TOMCAT outputs to investigate the impact of ENSO on the TCO over the TP.

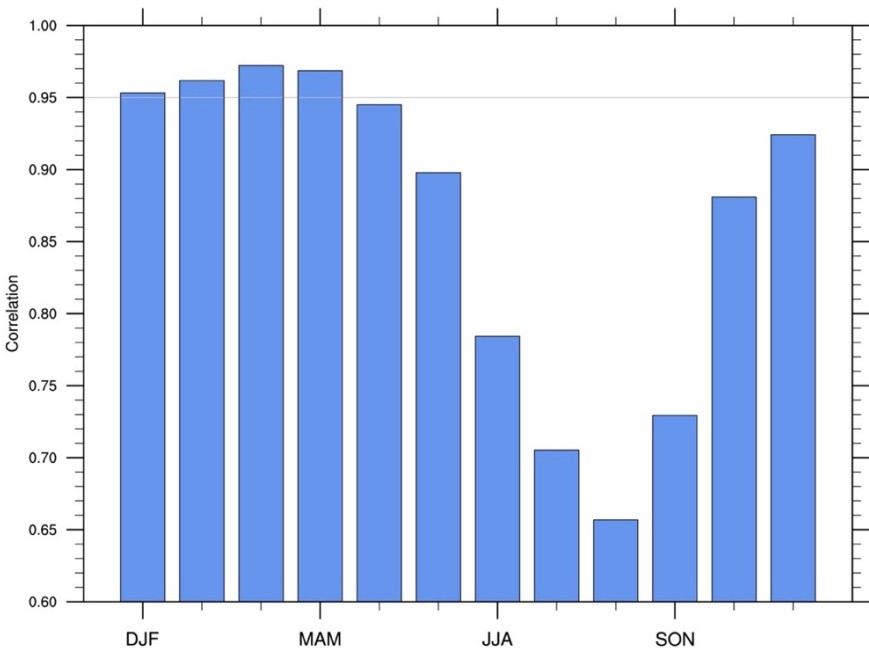

**Figure 1: Correlation coefficients between the time series of 3-month averaged TCO anomalies over the TP region (27.5–37.5°N, 75.5–105.5°E) for C3S dataset and TOMCAT simulation during 1979–2021. All correlation coefficients are statistically significant at 99% confidence level. The grey line represents the correlation coefficient equal to 0.95.**



### 3.1 Impacts of ENSO on the TCO over the TP

ENSO peaks in winter (DJF) and then decays in the following spring (MAM) and summer (June–July–August; JJA). To
investigate the impact of ENSO on the TCO over the TP, we now investigate the lag–lead correlation coefficients between the
TCO (C3S dataset: blue line; TOMCAT result: red line) and the winter (peak season, DJF averaged) Niño 3.4 index (**Figure
2**). Positive values indicate that ENSO is leading and a 3–month lead represents the MAM. Both the C3S dataset and TOMCAT
results show that the significant ENSO signal appears in the TCO before the ENSO's decay in May and the strongest ENSO
signal appears in the late winter/early spring (**Figure 2**). Comparing the TCO variance in both the C3S dataset and TOMCAT
results, it is observed that the variance is greater pre–May than post–May (**Figure 2**), indicating that ENSO may make more
contribution to the TCO variance pre–May. Overall, it can be seen from **Figure 2** that TOMCAT has a reasonable magnitude
of variance and correlation coefficients with ENSO from December. These results suggest that the significant response of TCO
over the TP to ENSO could extend from the December of ENSO's mature phase to the May of the decaying phase. Therefore,
the following composite TCO anomalies are averaged during December–May to maximize the signal, while we should note
the general relationship between ENSO and TCO is not sensitive to the chosen period as one could be expected from the
positive correlation during December–May.

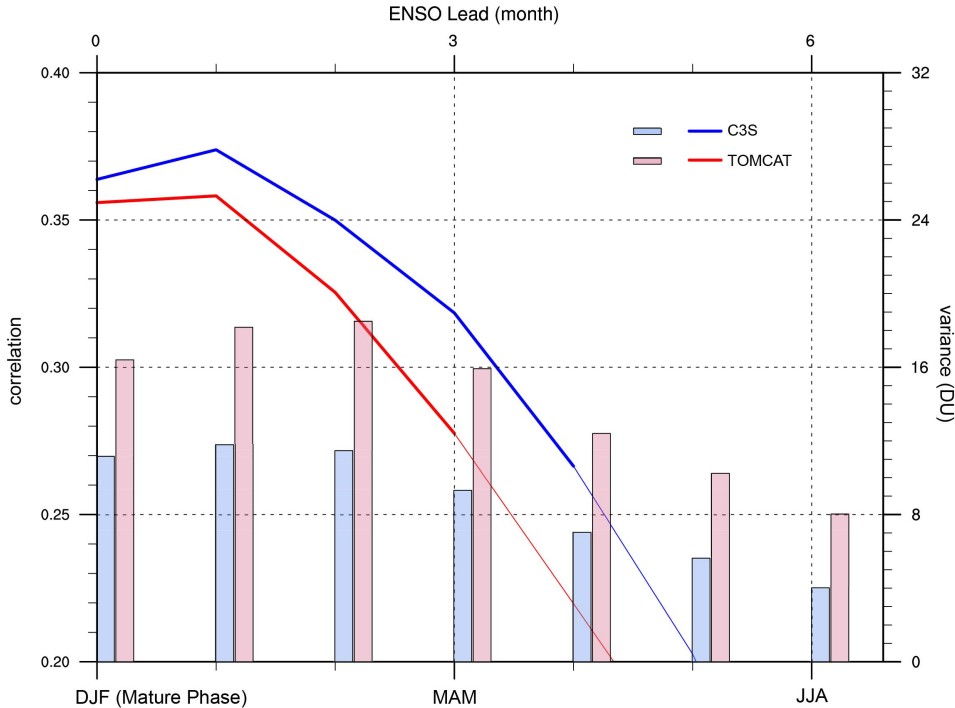

**Figure 2: Lead correlation between the time series of averaged TCO anomalies over the TP region (27.5–37.5°N, 75.5–105.5°E) and
winter (DJF) Niño 3.4 index for the period 1979–2021. Blue (red) line indicates C3S dataset (TOMCAT result). Positive leads indicate
that ENSO is leading. Lead values of 0, 3 and 6, indicate DJF, MAM, and JJA, respectively. Thicker lines indicate statistical
significance at the 99% confidence level. Bars (C3S: blue; TOMCAT: red) are standard deviations of TCO to measure its variance
(DU).**





Following the NOAA Climate Prediction Center, the criterion for such persistent El Niño and La Niña events is their mean of
195 anomalous Niño 3.4 index should be greater or less than 0.5 or –0.5 K, respectively, from DJF of ENSO's mature phase to
MAM of decaying phase. According to the criterion, the persistent El Niño and La Niña events from 1979 to 2021 are shown
in **Tables 1–2**. There are 7 and 9 events of the persistent El Niño and La Niña (**Tables 1–2**), respectively.

**Table 1: List of persistent El Niño events and their anomalous Niño 3.4 index (K) over the period 1979–2021. We identify persistent
events if their anomalous Niño 3.4 index is greater than 0.5 K from DJF to MAM (that is, DJF, JFM, FMA, MAM). The final row is
200 the number of total events and the last column is the corresponding mean of Niño 3.4 index.**

| El Niño | | | | | |
|---|---|---|---|---|---|
| Events | DJF | JFM | FMA | MAM | Mean |
| 1983 | 2.2 | 1.9 | 1.5 | 1.3 | 1.7 |
| 1987 | 1.2 | 1.2 | 1.1 | 0.9 | 1.1 |
| 1992 | 1.7 | 1.6 | 1.5 | 1.3 | 1.5 |
| 1998 | 2.2 | 1.9 | 1.4 | 1.0 | 1.6 |
| 2015 | 0.5 | 0.5 | 0.5 | 0.7 | 0.6 |
| 2016 | 2.5 | 2.1 | 1.6 | 0.9 | 1.8 |
| 2019 | 0.7 | 0.7 | 0.7 | 0.7 | 0.7 |
| Total Events: 7 | | | | | 1.3 |

**Table 2: Same as Table 1, but for La Niña events. The anomalous Niño 3.4 index (K) of these events is less than –0.5 K.**

| La Niña | | | | | |
|---|---|---|---|---|---|
| Events | DJF | JFM | FMA | MAM | Mean |
| 1985 | -1.0 | -0.8 | -0.8 | -0.8 | -0.9 |
| 1989 | -1.7 | -1.4 | -1.1 | -0.8 | -1.3 |
| 1999 | -1.5 | -1.3 | -1.1 | -1.0 | -1.2 |
| 2000 | -1.7 | -1.4 | -1.1 | -0.8 | -1.3 |
| 2008 | -1.6 | -1.5 | -1.3 | -1.0 | -1.4 |
| 2011 | -1.4 | -1.2 | -0.9 | -0.7 | -1.1 |
| 2012 | -0.9 | -0.7 | -0.6 | -0.5 | -0.7 |
| 2018 | -0.9 | -0.9 | -0.7 | -0.5 | -0.8 |
| 2021 | -1.0 | -0.9 | -0.8 | -0.7 | -0.9 |
| Total Events: 9 | | | | | -1.1 |

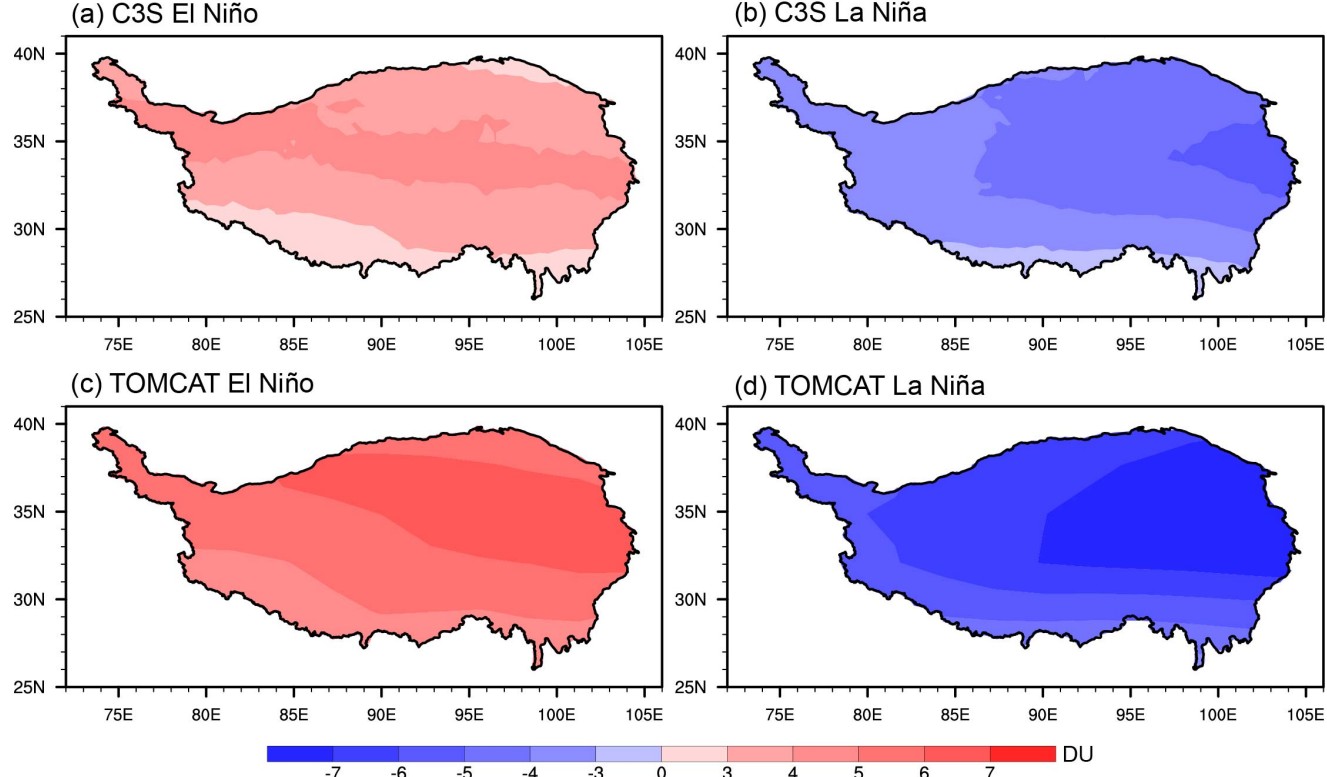

**Figure 3: Composite anomalies of the TCO (DU) for (a, c) El Niño and (b, d) La Niña events from the December of ENSO's mature phase to the May of decaying phase. Panels (a–b) are derived from the C3S dataset, and (c–d) are derived from TOMCAT results. Shaded regions indicate statistical significance at the 90% confidence level. The coloured region indicates the main body of the TP, and the black line represents the boundary of the TP. The dataset of the TP's boundary is from Zhang et al. (2002) and is available at https://www.geodoi.ac.cn/edoi.aspx?DOI=10.3974/geodb.2014.01.12.V1 (last access: July 2023).**

**Figure 3** shows composite anomalies of the TCO associated with the El Niño and La Niña events in **Tables 1–2** from the December of ENSO's mature phase to the May of decaying phase. For the El Niño (La Niña) events, there are significant positive (negative) TCO anomalies over the whole TP in both the C3S dataset and TOMCAT results (**Figure 3**). This result is consistent with that of correlation coefficients (**Figure 2**), highlighting the influence of ENSO on TCO over the TP. As the response of TCO over the TP to ENSO is consistent from December to May (**Figure 2**), it is worth noting that the composite results of TCO are not sensitive to the length of composite months or seasons during the December–May. On the whole, both the C3S measurement-based dataset and TOMCAT model results show that ENSO has an effect on the TCO over the TP. To be specific, it is a prolonged impact from the December of the ENSO's mature phase to the May of the decaying phase, and the El Niño (La Niña) events induce the positive (negative) TCO anomalies. To better understanding the effect of ENSO on the TCO, it is necessary to investigate the vertical ozone changes associated with ENSO.



## 3.2 Impact of ENSO on the ozone profile

**Figure 4** shows composite percentage change (i.e., anomaly divided by climate mean) of the ozone profiles for the El Niño and La Niña events. To make a direct and convenient comparison, here we interpolate the TOMCAT model level (sigma–

pressure coordinate) results to the same SWOOSH vertical resolution (pure pressure level). From the SWOOSH dataset and TOMCAT results, **Figure 4** depicts that the El Niño (La Niña) events mainly affect the level of ozone at 200–70 hPa, where there are significant ENSO signals. It can also be clearly seen from **Figure 4** that the El Niño (La Niña) events can induce the remarkable increase (decrease) of ozone at 200–70 hPa for the SWOOSH dataset and TOMCAT results. These changes further contribute to the positive (negative) TCO anomalies over the TP. Although the TOMCAT results at 200–70 hPa overestimate

the absolute percentage change of ozone profile for the El Niño and La Niña events compared to the SWOOSH dataset (**Figure 4a**), the TOMCAT and SWOOSH dataset exhibit similar composite results.

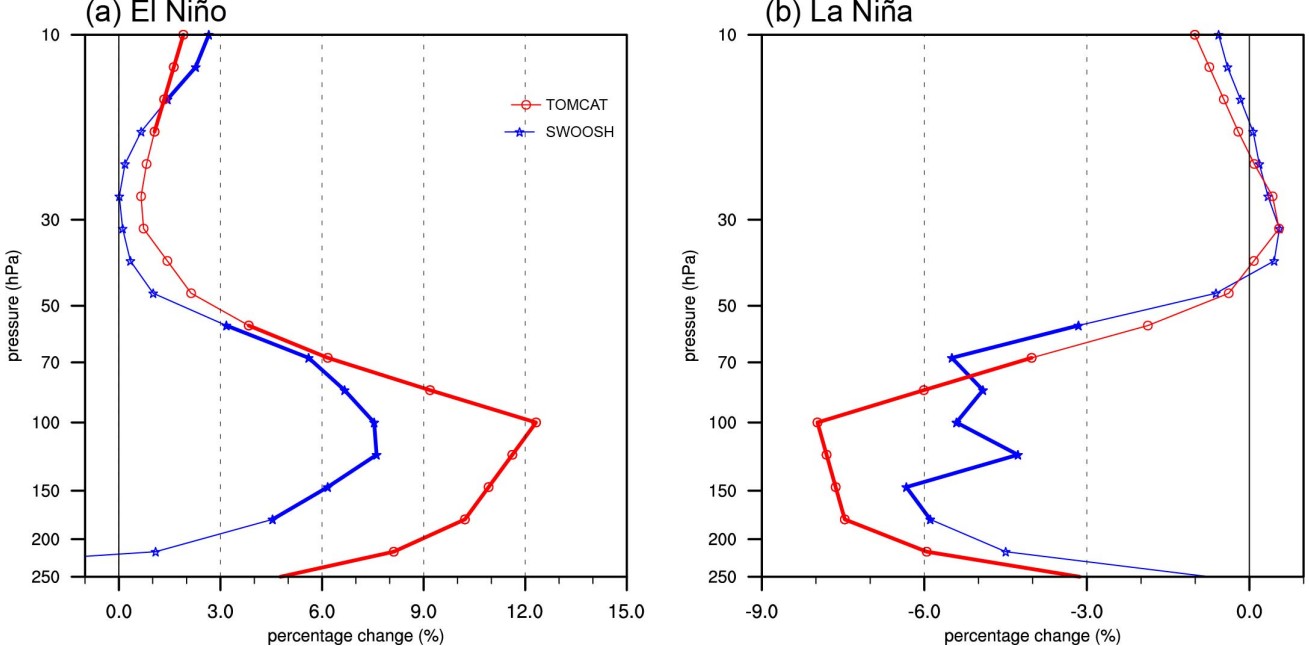

**Figure 4: Composite percentage change (%) of the ozone profiles for (a) El Niño and (b) La Niña over the TP region (27.5–37.5°N,**
**75.5–105.5°E) from the December of ENSO's mature phase to the May of decaying phase. Red lines are derived from TOMCAT, and blue lines are derived from the SWOOSH dataset; thick lines indicate values which exceed the 90% confidence level.**



Figure 5: Composites and climate means of the ozone profiles from (a) the TOMCAT result and (d) SWOOSH dataset over the TP region (27.5–37.5°N, 75.5–105.5°E) from the December of ENSO's mature phase to the May of decaying phase. (b-c) as in (a), but for a zoom on the 200–100hPa and 100–50hPa domains of (a), respectively; (e-f) as in (b-c), but for the TOMCAT result. Green lines indicate climate means of ozone profiles, red lines indicate composite ozone profiles of El Niño events, and blue lines indicate composite ozone profiles of La Niña events.






As the main level of the El Niño (La Niña) influence on the ozone profile is centered at 200–70 hPa (**Figure 4**), it is important to understand how the El Niño (La Niña) events cause ozone profile anomalies in this height domain. **Figures 5a** and **5d** shows the climate mean (green line) and composite ozone profiles for the El Niño (red line) and La Niña (blue line) events from 200 hPa to 50 hPa. To amplify the differences between the composite ozone profiles and climate mean, we zoom in on the ozone
profile at 200–100 hPa (**Figures 5b** and **5e**) and 100–50 hPa (**Figures 5c** and **5f**). At the main impact levels of ENSO (i.e., 200–70 hPa) in **Figure 4**, we note that the El Niño (La Niña) events induce downward (upward) shift of ozone profile compared to the climate mean in both the SWOOSH dataset and TOMCAT results (**Figure 5**), which can further cause the positive (negative) ozone anomalies (**Figure 4**) at 200–70 hPa and hence contribute to the positive (negative) TCO anomalies over the TP (**Figure 3**).


### 3.3 Potential mechanism for the impact of ENSO on the TCO

Since tropopause variability has a large impact on the ozone profile and TCO (e.g. Schubert and Munteanu, 1988; Steinbrecht et al., 1998; Varotsos et al., 2004; Tian et al., 2008; Zhang et al., 2014), **Figures 6a**–**6b** show composite anomalies of the tropopause pressure for the El Niño and La Niña events. The El Niño (La Niña) events generally correspond to a positive
(negative) tropopause pressure over almost the whole TP, where the significant anomalies are located broadly between 30° to 35°N (**Figures 6a**–**6b**). These results indicate that the El Niño (La Niña) events can induce a sinking (lifting) tropopause height (TH) above the TP. **Figures 6c**–**6d** show latitude–height section of the composite partial column ozone anomalies and tropopause height. During the El Niño events (**Figure 6c**), the sinking TH (green line) compared to its climate mean (purple line) corresponds to the significantly positive anomalies of partial column ozone at about 200–70 hPa, which is consistent with
composite percentage change of the ozone profile associated with El Niño events in **Figure 4a**. Such positive partial column ozone anomalies related to El Niño events further results to the positive TCO anomalies (**Figures 3a** and **3c**). The La Niña events corresponds to opposite sign in comparison to the El Niño events (**Figure 6**). Specifically, the lifting TH (green line in **Figure 6d**) relative to its mean (purple line) favours the negative partial column ozone, further contributing to the negative TCO anomalies during the La Niña events (**Figures 3b** and **3d**). These results indicate that the TH plays a vital role in the
TCO variation of the TP during the El Niño and La Niña events.





**Figure 6: Composite anomalies of tropopause pressure (hPa) for (a) El Niño and (b) La Niña events from ERA5 dataset over the TP. Latitude–height section of the composite partial column ozone anomalies (DU, averaged from 75.5°E to 105.5°E) for (c) El Niño and (d) La Niña from TOMCAT. All variables are averaged from the December of ENSO's mature phase to the May of decaying phase. Stippled regions indicate statistical significance at the 90% confidence level. Purple lines in (c–d) indicate the climate mean of tropopause, and green lines indicate the composite tropopause for (c) El Niño and (d) La Niña, respectively. The grey shading indicates the topography.**



Given that the TH is closely related to atmospheric circulation and the SSTA associated with ENSO plays a vital role in atmospheric circulation, **Figure 7a–7b** show composites of SSTA from the December of ENSO's mature phase to the May of decaying phase. It can be clearly seen the maximum SSTA in the central and eastern tropical Pacific for the El Niño and La Niña events. Aside from the signal over the tropical Pacific, there is also a significant signal over the tropical Indian Ocean, where the El Niño (La Niña) events correspond to a basin–wide warming (cooling) SSTA. Previous studies have demonstrated that the basin–wide warming (cooling) over the tropical Indian Ocean is a response to surface heat flux changes induced by El Niño (La Niña) (e.g. Alexander et al., 2002; Lau and Nath, 2003; Yang et al., 2007; Schott et al., 2009).

As the geopotential height of upper–level atmospheric circulation at 150 hPa (GH150) is considered as an important proxy of atmospheric circulation over the TP (see Li et al., 2020), **Figure 7c–7d** display its composites for the El Niño (La Niña) events derived from ERA5 dataset from the December of ENSO's mature phase to the May of decaying phase. As shown in **Figure 7c**, there are significant positive GH150 anomalies over the tropical Pacific and Indian Ocean and negative GH150 anomalies over the TP for the El Niño events. The GH150 anomalies associated with the La Niña events (**Figure 7d**) are generally opposite to the El Niño events (**Figure 7c**). Such patterns of **Figure 7c–7d** are similar to previous studies (e.g. Matsuno, 1966; Gill, 1980; Wallace and Gutzler, 1981; Jin and Hoskins, 1995; Xie et al., 2016), which have demonstrated that the anomalous geopotential height of upper–level atmospheric circulation associated with ENSO is a response of SSTA over the tropical Pacific and Indian Ocean (**Figure 7a–7b**). Particularly, the positive (negative) GH150 anomalies over the tropical Pacific and Indian Ocean (**Figure 7c–7d**) are induced by the El Niño (La Niña) events (**Figure 7a–7b**), because the warming (cooling) SSTA over the tropical Pacific and Indian Ocean (**Figure 7a–7b**) can increase (decrease) tropospheric temperature and thus elevate (depress) upper troposphere geopotential height (e.g. Xie et al., 2009; Dhame et al., 2020). Following previous studies, it is suggested that ENSO is responsible for anomalous geopotential height over the TP through two pathways, one of which is tropical Pacific SSTA (**Figure 7a–7b**) via modifying the stationary planetary wave along the westerly jet stream (e.g. Trenberth et al., 1998; Zhang et al., 2015b), and the other of which is the basin–wide SSTA over the tropical Indian Ocean (**Figure 7a–7b**) via atmospheric wave response (Jin and Hoskins, 1995) or via modifying the land–sea thermal contrast between the tropical Indian Ocean and TP (e.g. Chen and You, 2017; Zhao et al., 2018). Whichever pathway is operating, the GH150 anomaly over the TP has been suggested to be associated with ENSO because the basin–wide SSTA over the tropical Indian Ocean (**Figure 7a–7b**) is a response of ENSO (e.g. Alexander et al., 2002; Lau and Nath, 2003; Yang et al., 2007; Schott et al., 2009). The above results highlight the key role of ENSO in the upper–level geopotential height over the TP. That is, the El Niño (La Niña) events induce negative (positive) anomalies of upper–level geopotential height.





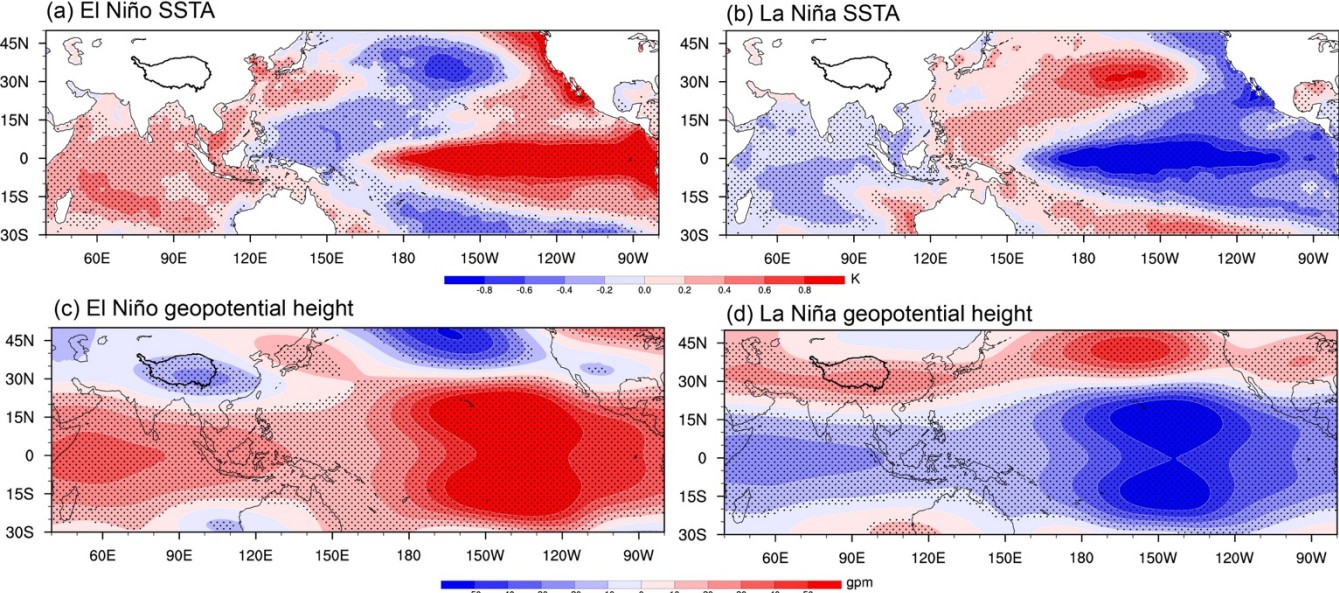

**Figure 7: Composites of SSTA (K) for (a) El Niño and (b) La Niña from HadISST1 dataset from the December of ENSO's mature phase to the May of decaying phase. (c–d) as in (a–b), but for geopotential height (gpm) anomalies at 150 hPa from ERA5 dataset. Stippled regions indicate statistical significance at the 90% confidence level. The black lines represent the boundary of the TP.**

Based on equation (3), such anomalous upper–level geopotential height associated with ENSO can induce atmospheric thickness anomalies and thereby influence the tropospheric air temperature. Over the TP, we calculated the correlation coefficient between monthly time series of tropospheric mean temperature and temperature estimated via the layer from 700 to 150 hPa thickness. We find their correlation coefficient is close to 1.0, indicating that tropospheric mean temperature is closely related to the air thickness. That is, the tropospheric temperature will warm (cool) when the rising (falling) of upper–level geopotential height causes the increased (decreased) air column thickness. Although the TH can be influenced by both stratospheric and tropospheric processes, here we show that the TH over the TP associated with ENSO is dominated by the tropospheric air thickness. We will address this finding by tracing the ENSO signal to the tropospheric air thickness and associated tropospheric temperature. **Figures 8a–8b** shows map of the composites of temperature associated with air thickness. Significantly negative temperature anomalies associated with thickness occur during the El Niño events (**Figure 8a**), and vice versa for La Niña (**Figure 8b**). It is not surprising that these composite pattern and magnitude (**Figures 8a–8b**) are generally the same as that of tropospheric mean temperature (**Figures 8c–8d**) because of their close relationship. According to equation (3), this implies that the El Niño (La Niña) events favour the decreased (increased) air thickness and thus cause cooling (warming) tropospheric mean temperature (**Figures 8**).

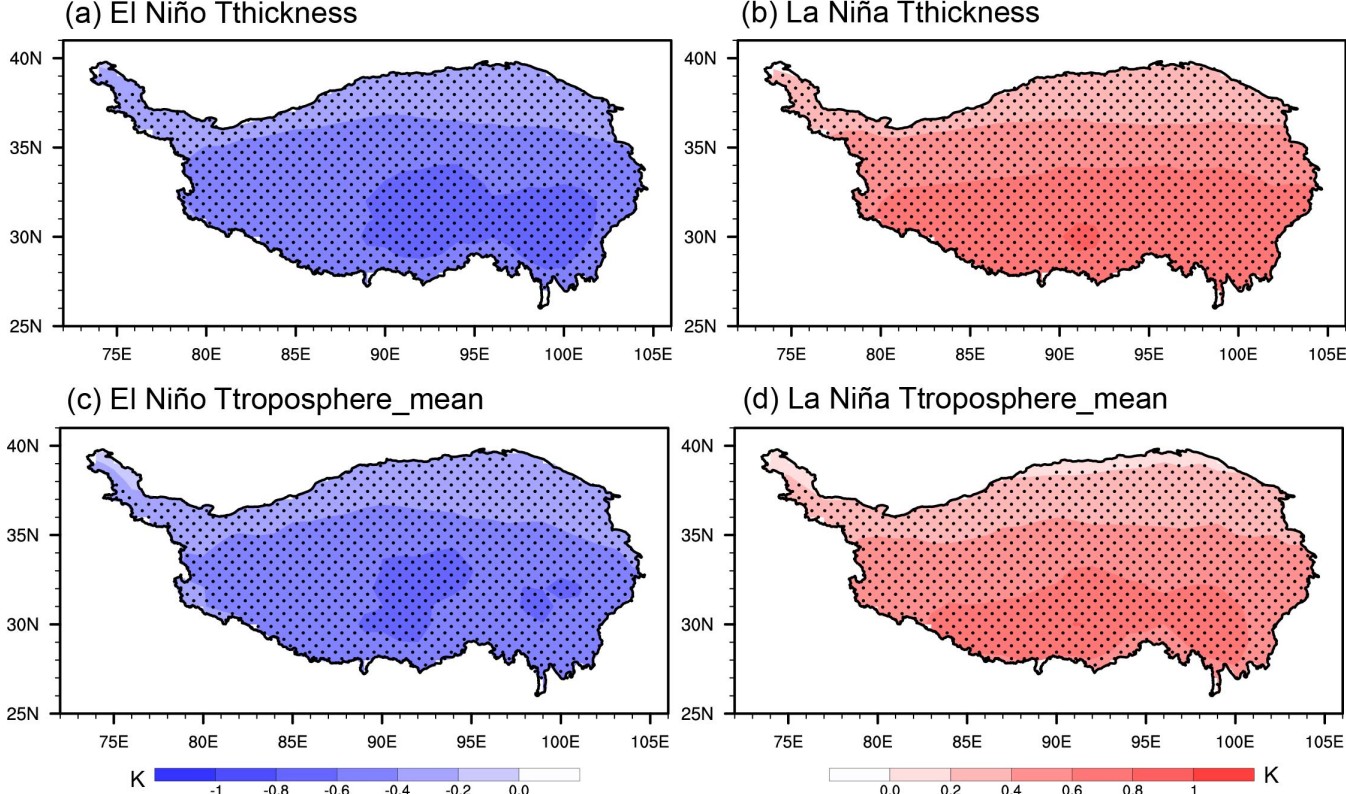

Figure 8: Composite anomalies of the mean temperature (K) anomalies estimated via the layer (700–150 hPa) thickness (Tthickness) for (a) El Niño and (b) La Niña events derived from ERA5 dataset from the December of ENSO's mature phase to the May of decaying phase. (c–d) as in (a–b), but for the tropospheric mean temperature (K) anomalies (Ttroposphere_mean). Note that we calculate the tropospheric mean temperature from 700 hPa to 150 hPa because of the altitude of the TP (about 700 hPa) and the mean TH (about 150 hPa). Stippled regions indicate statistical significance at the 90% confidence level. The black lines represent the boundary of the TP.

To further show the relationship between ENSO and the tropospheric air thickness and temperature, we plotted in Figure 9a the temperature associated with air thickness as a function of the ENSO index (Niño 3.4). From the monthly scatter diagram, there is a strong negative correlation (–0.56) between Niño 3.4 and temperature associated with air thickness, with significance above the 99% confidence level. This result indicates that the cooling (warming) temperature associated with air thickness is closely related to the El Niño (La Niña) events (**Figure 9a**), which is in good agreement with the composite results in **Figure 8**. In addition, **Figure 9b** shows the monthly TH anomalies as a function of the temperature associated with air thickness for ENSO events. The significantly strong negative correlation (–0.82) between them implies that TH over the TP associated with ENSO is dominated by the tropospheric air thickness. The good separation between El Niño and La Niña groups are coherent with composite results, with El Niño (La Niña) corresponding to a cooler (warmer) tropospheric temperature and a lower (higher) TH. In sum, these results suggest that the El Niño (La Niña) events can modulate upper–level geopotential height



(**Figure 7**), induce thinning (thickening) atmospheric thickness over the TP (**Figures 8**), cause the cooling (warming) tropospheric temperature (**Figure 8**), and finally lead to the sinking (lifting) of TH (**Figure 9b**).

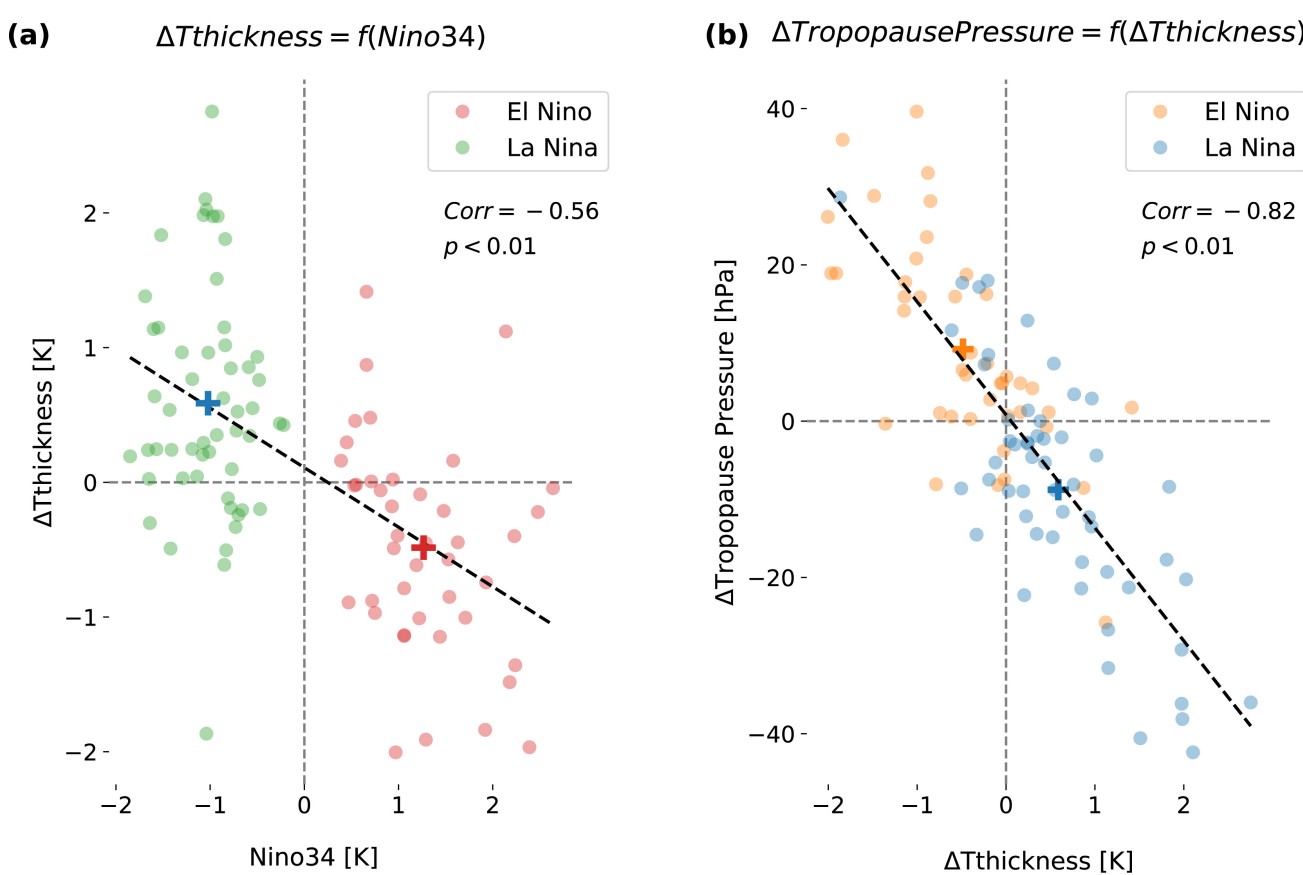

**Figure 9: (a) The temperature associated with air thickness (Tthickness, unit: K) as function of Niño 3.4 (K) for El Niño (red circles) and La Niña (green circles) events. (b) The tropopause pressure (hPa) as function of Tthickness for El Niño (orange circles) and La Niña (blue circles) events. The averaged values for El Niño and La Niña events are shown as pluses, and the linear regression lines are shown as dashed black lines.**

**4 Summary and discussion**

This study aims to investigate the influence of ENSO on the TCO over the TP and its associated potential dynamical mechanism by using merged satellite, TOMCAT model output, ERA5 reanalysis, and HadISST1 datasets. Our results show that the correlation coefficient between the monthly TP's TCO time series of TOMCAT results and the merged satellite TCO from C3S dataset is strong (about 0.92) over the period 1979–2021 and that the correlation coefficient is statistically significant at the 99% level based on the two–tailed Student's $t$–test and the $N^{\text{eff}}$ of degrees of freedom. In particular, for the DJF, JFM,

FMA, and MAM, the TOMCAT results and C3S data are highly correlated (above 0.95, **Figure 1**), indicating that the TCO





changes over the TP in the TOMCAT and C3S dataset are very similar from December to May. Therefore, we use them to investigate the impact of ENSO on the TCO over the TP.

Through analysis of lead correlation between ENSO and TCO over the TP, both C3S dataset and TOMCAT results show that ENSO has a significant impact on the TCO from the December of its mature phase to the May of decaying phase (**Figure 2**). In its positive phase (El Niño) events from December to May, it can induce positive TCO anomalies over the TP, with conditions reversed for the negative phase (La Niña) events (**Figure 3**). The TCO variability associated with the El Niño and La Niña events is closely related to the ozone profile change. The SWOOSH dataset and TOMCAT results show that the El Niño and La Niña events mainly influence the level of ozone centered at 200–70 hPa (**Figure 4**). This is because the El Niño (La Niña) events induce downward (upward) shift of ozone profile compared to the climate mean (**Figure 5**) and therefore contribute to the positive (negative) ozone anomalies (**Figures 3–4**).

We highlight the potential mechanism for the impact of ENSO on the TCO over the TP from December to May. The El Niño (La Niña) events can induce the falling (rising) upper–level geopotential height (**Figure 7**) and thus lead to the decreasing (increasing) air thickness (**Figures 8a–8b**), which in turn causes the cooling (warming) tropospheric temperature over the TP (**Figures 8c–8d**). Since the TH over the TP associated with ENSO is dominated by the tropospheric temperature (**Figure 9**), these results indicate that the El Niño (La Niña) events could cause a decrease (increase) of the TH (**Figure 6**), which contributes to the downward (upward) shift of ozone profile (**Figure 5**) and hence induce the TCO increase (decrease) (**Figure 3**). Our results suggest the El Niño (La Niña) events play an important role in the TCO variability over the TP. Recently, climate models project the increasing frequency of strong El Niño (La Niña) events due to greenhouse–gas–warming forcing (e.g. Cai et al., 2018). This indicates that the El Niño (La Niña) events may have a greater and stronger impacts on the TP's ozone in the future under greenhouse–gas–warming compared to the present and past.

In this study, we provided a systematic explanation to the impacts of ENSO on the TCO over the TP via the TH. Although **Figure 9** is in good agreement with our study and shows that there are significant correlations between monthly samples of ENSO events and air thickness as well as TH, it is also observed a few samples deviate from the regression line. This implies that in addition to ENSO, there may be other factors resulting in the air thickness and TH variability and thus contributing to the TCO variation. Aside from ENSO, it is worth noting that there are many factors influencing the TCO over the TP, for example, convective activities (Liu et al., 2003), anticyclone in the upper troposphere (Zheng et al., 2004), Asian summer monsoon (Bian et al., 2011), South Asian high (Guo et al., 2012), and QBO (Chang et al., 2022). Recently, Duan et al (2023) stated that the tropical Indian Ocean SSTA could cause a vertical shift of ozone profile over the TP and then contribute to the TCO variation. Their study is different from ours, as we focus on ENSO with the strongest interannual SSTA. Considering the close relationship between ENSO and tropical Indian and Atlantic Oceans, it will be interesting to study their individual and combined effects on the TP's TCO. In addition to the tropical SSTA, the interaction of different processes mentioned above and their contributions to the TCO over the TP are fundamental topics, requiring further study.



*Data availability.* The satellite and climate data used in this study are available at the source and references given in Section 2. The model output used for the figures are available at https://doi.org/10.5281/zenodo.7995860 (Li et al., 2023).

*Author contributions.* YL performed the data analysis and prepared the manuscript. WF, MPC, XZ and YL gave support for discussion, simulation, and interpretation, and helped to write the paper.

*Competing interests.* The authors declare that they have no conflicts of interest.

*Acknowledgements.* The authors thank the Copernicus Climate Change Service (C3S), ECMWF, NOAA, and Met Office Hadley Centre for providing data. The modelling work is supported by University of Leeds and National Centre for Atmospheric Science (NCAS). This work was jointly supported by the National Natural Science Foundation of China (grant nos. 42175042, 42275059, and U20A2097), the China Scholarship Council (grant nos. 201908510031 and 201908510032), and Natural Science Foundation of Sichuan Province (grant nos. 2022NSFSC1056 and 2023NSFSC0246). The TOMCAT modelling work was also supported by the Natural Environment Research Council (NERC) (grant no. NE/V011863/1).

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
