# Peer review of "The impact of El Niño–Southern Oscillation on the total column ozone over the Tibetan Plateau"

_EGUsphere, 2023_

## Author Comment (AC1)

**Response to the Referee #1's comments on "The impact of El Niño–Southern Oscillation on the total column ozone over the Tibetan Plateau" (egusphere-2023-1452)**

We thank the Referee #1 for making detailed comments and very useful suggestions to improve the paper. The manuscript has been revised and improved in response to the referee's comments and suggestions. Below is a point-by-point response (in black) to the referee's comments (in blue) followed by any modifications to the manuscript (*in italics*). We have added a new figure to show the great coherence between the ozone changes and the tropopause changes in terms of their spatial patterns. We have also included a caveat in discussion on the uncertainties of the results.

The line numbers for the changes correspond to the clean/revised manuscript version.

Response to *Anonymous Referee #1*

*General comments*

This article aims to explore the impact of ENSO on the variability of the total ozone column over the Tibetan Plateau. The topic of the work is absolutely interesting, as the subject is not much investigated and the mechanisms driving the variability of ozone over the Tibetan Plateau are not fully elucidated. Also, the paper is well written, so below there are only a limited number of technical corrections to improve the readability.

Many thanks for the positive evaluation and the suggestions to improve the paper.

However, the paper suffers from many important drawbacks in the methodology applied, and therefore calls for major improvements before it can be accepted for publication on Atmospheric Chemistry and Physics.

First of all, the analysis is mostly based on the use of correlation coefficients, which alone cannot be used neither to document the performance of model simulations against observations, neither to prove the existence of physical links between two (or more) atmospheric processes. Indeed, the presence of a correlation in two variables is not alone sufficient to claim the existence of a cause-effect relationship. Even more so as in this case the analysed variables present well different spatial patterns (which is never clearly discussed).

Thank you for the comments and concern. For the performance of model simulations against observations, the high correlation (above 0.95, Figure 1) of TP TCO between C3S dataset and TOMCAT simulation from December to May give us confidence that the TOMCAT is able to capture the observed variability in TP TCO during these seasons and that we can thus use it to investigate the impact of ENSO on the TP TCO change.

For the correlation analysis, we agree with the reviewer that the correlation analysis has its limitation. We apologise for the lack of discussion about the spatial pattern. We have now added a caveat in summary and discussion [Lines 406–410], and added a plot about partial column ozone and discussion showing their great coherence in spatial patterns [see Figure 6; Lines 280–283]. The composite analysis from Figure 6 highlights the coherence between the ozone changes and the tropopause changes during ENSO events, which further supports our correlation results.

[Lines 406–410]: "*Our study focuses on the diagnosed ozone changes over the TP during ENSO episodes using both observations and a chemistry transport model TOMCAT as well as several statistical methods, which will have some uncertainties due to large internal variability of ozone and limited ENSO events. Future work is needed for a better understanding of tangible ENSO impacts with more observed ENSO events and a full-chemistry climate model*".

[Lines 280–283]: "*Considering that the area–averaged climate mean of TH over the whole TP is about 150 hPa during December–May, Figures 6c–6d show the composite anomalies of the partial column ozone at 150 hPa. Their spatial patterns (Figures 6c–6d) are in good agreement with the composite TH anomalies (Figures 6a–6b), highlighting the good coherence between the ozone changes and the tropopause changes*".

Secondly, the authors focus the analysis on composite averages analyzing simultaneously multiple El Niño (La Niña) events, and it is not clear if the results can be ascribed to all events or just to some major leading ones. Tropopause height, but also ozone, present relevant day to day changes, which have not been thoroughly analysed and documented here.

The composite results are statistically significant according to the two–tailed Student's $t$–test, ruling out the possibility that the results are dominated by a single event. This is also clear by looking at the scatter plot (Figure 9) with each event represented as a dot showing that the relationship between ENSO index and ozone-related changes is evident with most of the events coherent with composited results.

For the second part of the comment, the reviewer suggests a day-to-day analysis, which will be helpful to take account of the seasonal resolved impacts or subseasonal changes. However, this is out of scope of this study.

Thirdly, the reanalysis, satellite measurements and model analysis utilized have very different (and coarse) resolution, which can lead to significant drawbacks in the analysis. On the basis of my review, I guess that the paper would benefit from a major revision and explanation of the methodology applied and from an improved discussion to motivate, interpret and prove the results obtained. Something also in the direction of the suggestions from the editor, highlighting that other modes of variability can impact on the total ozone time series, can be also beneficial.

Thank you for this. Yes, the reanalysis, satellite measurements and model analysis have different resolutions in the horizontal and vertical. The current work mainly focuses on the Tibetan Plateau region (27.5–37.5°N, 75.5–105.5°E), so these data have been weight-averaged to the same region which is mentioned in the paper (Figures 1, 2, 4, 5 and 9). For the region results, we use the available reanalysis product and satellite dataset which have been properly cited in their descriptions.

For the TOMCAT model simulation, we use the standard setup (with 2.8° horizontal resolution) for the long-term simulation (1979-2021), which is able to capture the general features of ozone and other chemical species. Previous studies (Feng et al., 2005a, 2011, Chipperfield et al., 2005, 2006, thereafter) showed that different horizontal resolutions would make some changes in the tracer transport and modeled tracers distribution, there is a significant improvement when using the resolution 2.8° compared to 7.5° (e.g., Chipperfield et al., 2005, Feng et al., 2005b), but slight improvement when using 1-degree resolution (T106 Gaussian grid) (Feng et al., 2011). Overall, the 2.8° is still reasonable well able to capture the general features of the simulated ozone and other tracers when compared with different measurements.

*Specific comments*

1. Lines 14-15: Well, more precisely, the solar ultraviolet radiation and the derived risks to human and ecosystems health are controlled by the stratospheric ozone. It is true that the total ozone column is essentially equal to the stratospheric content as the tropospheric ozone concentration is too low as compared to the stratospheric one, but the sentence should be more precise.

We have modified our sentence as noted: [Line 15] *"……which is mainly controlled by the local ozone in the stratosphere*".

2. Lines 15-18 and 62-64: ENSO has both an oceanic (El Nino) and an atmospheric component (Southern Oscillation), but here you refer only to the oceanic one. Please describe better.

Revised.
[Lines 15-18] "*ENSO……is characterized by the tropical Pacific Ocean sea surface temperature anomalies (SSTA) and sea level pressure change for the warm phase El Niño and cold phase La Niña events.*"

3. Lines 28-32: The explanation is quite confused and not straightforward, I would suggest to rephrase to make it clearer.

Revised.
[Lines 25-32] "*…… This reduced temperature associated with El Niño events causes a decrease of the tropopause height, which tends to replace ozone–poor tropospheric air by ozone–rich stratospheric air in the UTLS and hence leads to the increase in TCO*".

4. Lines 32-36: I do not understand the meaning of "descending upper-level geopotential height". Explain better. Also I suggest rephrasing: "Our results suggest that the El Niño events lead to a

descending upper–level geopotential height and hence cause a decrease in air column thickness, which in turn induces reduced tropospheric temperature over the TP."

The "descending upper–level geopotential height" means a "*negative upper–level geopotential height anomaly*", which has been explained in revised manuscript [Line 27].
Thank you for the suggestion, we also have rephrased our sentence [Lines 26-28].

5. Lines 34-36: I do not understand how just a shift in the ozone profile can lead to an increase in TCO.

The shift of ozone profile represents the partial column ozone anomalies associated with TH change. We have updated explanation for clarification, please also see the response to point 3.
[Lines 272-276]: "*Approximately 90% of ozone in the atmospheric column resides in stratosphere; the ozone concentration is much lower in the troposphere with a gradual transition at the tropopause. Therefore, a decrease of tropopause height (TH) will tend to replace ozone–poor tropospheric air by ozone–rich stratospheric air in the UTLS region, and thus increase the partial column ozone, which in turn contributes to the TCO increase, and vice versa for an increase of TH (e.g. Schubert and Munteanu, 1988; Salby and Callaghan, 1993; Steinbrecht et al., 1998; Chipperfield et al., 2003; Varotsos et al., 2004; Tian et al., 2007)*".

6. Lines 38-39: And what about the frequency of the events? If the events of the two kinds change intensity of the same magnitude, but with opposite effects on ozone, wouldn't be the final change almost null? Please explain better.

The frequency of strong ENSO events (El Niño and La Niña) will be increasing under greenhouse–gas–forced warming (e.g. Cai et al., 2015, 2018). However, it is still uncertain the magnitude intensity of events and which phase events (El Niño or La Niña) is increasing, owing to the lack of inter–model consensus (e.g. Cai et al., 2015, 2018; Collins et al., 2010). We have modified our sentence as noted [Line 33]: "*…… climate models project an increase in the frequency of strong El Niño or La Niña events*".

7. Lines 12-41: The abstract is too long and reporting many results of the correlation tests, while it is better to focus on the interpretation of the results and the underlying mechanisms. Please revise.

We have removed some details on the correlation tests and highlighted the impact of ENSO on the TP TCO and its underlying mechanisms [Lines 13-35].

8. Line 47: Please explain better how the elevation of the site is linked to the low air density and high atmospheric transparency.

As altitude increases, the number of gas molecules in the air decreases, and therefore the air becomes less dense (Ahrens and Samson, 2011). We also changed "atmospheric transparency" to "*clean air (Pokharel et al., 2019)*". These citations have been added [Line 41].

9. Lines 48-49: How is this connected with the previous sentence?

The connection is: the presence of ozone is crucial for any life because the UV radiation is harmful to the biota. Please see [Lines 41-45].

10. Lines 54-58: Those mechanisms that you listed here are various and of different nature. I would suggest better description.

We have improved these sentences by classifying these proposed mechanisms into two different processes, with one on the stratospheric process and another on tropospheric changes. The revised sentence [Lines 48-52] is also shown below.
"*These studies have argued that summertime TCO low is caused by changes in mass exchange between troposphere and stratosphere due to the stratospheric variability, for example the synchronisation of the quasi–biennial oscillation (QBO) and seasonal cycle (Chang et al., 2022), and tropospheric changes, for example the high topography and thermal forcing of the TP (Ye and Xu, 2003; Kiss et al., 2007; Tian et al., 2008; Guo et al., 2012) and enhanced convective activity in summer (Liu et al., 2003; Bian et al., 2011).*"

11. Lines 58-60: Before you were talking just about summer TOL, now you talk about winter and spring events, perhaps (not clear) connected to different processes. Revise.

We have revised it to make better connection, as noted [Lines 52-55]: "*In comparison to the summertime TCO change, less attention was paid to the TP TCO variability for other seasons. It is worth highlighting that the interannual variability of TP TCO from wintertime to springtime is strongest (Figure S1 of the Supplement)*".

12. Lines 61-62: As commented above, the link between QBO and TOL is not explained. Also, in this sentence, it is not clear if there is a link between QBO and ENSO. Revise.

We have added the link between QBO and TCO low in the revised manuscript [Lines 55-56]. We also have added a link between QBO and ENSO [Line 186].

13. Lines 64-66: Are you talking about climate or about meteorology or of the Earth system when you talk about the interannual variability of ENSO?

We are talking about climate. This is now specified.

14. Lines 68-69: The sentence and the reason why most studies have focused on the polar regions and the tropical stratosphere is not clear.

We have now made it clear [Lines 65-66]: "……*considering the major ozone production in the tropics and ozone depletion in the polar region (e.g. Staehelin et al., 2001)*".

15. Line 72: I would not label 1979-2002 measurements as "very limited", if not explained better what the limitations are.

As the period 1979–1992 (14 years) used by Zou et al. (2001) is relatively short and the ENSO events (3 El Niño, 2 La Niña) are too few based on Tables 1–2, we feel that their results are based on very limited observations. Added it as noted [Lines 68-69]: "*However, their results are based on very limited ENSO events since the satellite era from 1979 to 1992*".

16. Line 74: You never talked of a limitation in spatial coverage, so the reader has no way to compare the claimed much wider range of your work against previous ones.

Revised.
[Lines 71-72]: "……*we use a longer period of ozone measurements over the period 1979–2021*".

17. Line 75: It is not exactly clear, as stated now, what the chemical transport model adds to the analysis.

We have added it as noted [Lines 72-73]: "*along with the TOMCAT chemical transport model simulations (e.g., Chipperfield et al., 2017, 2018)*".

18. Lines 81-83: This sentence is more appropriate for the conclusion section rather than for the introduction.

Thank you for this. We have deleted this sentence in the revision.

19. Lines 102: The resolution seems to be rather coarse, which can pose limitations to the study.

Please see the detailed response mentioned above (3[rd] *General comment*).

20. Lines 86-119: I would suggest explaining better what kind of measurements/observations are derived from each source, as the sources are many and of different kinds: satellite, reanalysis, …

Thank you for this. The source has been labeled in revised manuscript [Lines 80-116].

21. Line 128: Usually ECMWF means that you are not using always ECMWF data? From what does this depends? Also, which ECMWF reanalysis?

All chemical transport models (CTMs) require the forcing files to determine the atmospheric background (winds, temperature, humidity). It also applies to the TOMCAT CTM which is forced by meteorological dataset, usually ECMWF and sometimes UK Met Office (UKMO; Swinbank and O'Neill, 1994). Both show that TOMCAT produces a good simulation compared to the observations (e.g. Feng et al., 2003). In the current study, we have used the latest ECMWF reanalysis product (ERA5) because of its longer coverage period to ensure the consistency (ERA-Interim stopped in August 2019, Li et al., 2020). Feng et al. (2007) noted some abrupt changes in temperature when moving one ECMWF product to another for the long-term simulation (ERA40 to ERA-Interim) and argued the unrealistic variations in the analyses used to force the model. In any case, to make our sentence clearer, we have deleted "*(usually ECMWF)*" in revised manuscript.

22. Lines 127-129: The sentence is not clear, as presented now, as it repeats that the model is forced with ECMWF winds and temperatures to specify atmospheric transport and temperatures. Revise.

We have revised repeated sentences [Lines 124-127].

23. Lines 130-131: Yet another different, and coarse, resolution...

Please see the detailed response mentioned above (3rd *General comment*).

24. Lines 135-136: This also depends on the purpose of the investigation...

We have deleted the judgement on different definitions [Lines 130-131].

25. Lines 157-168: The correlation coefficients alone cannot prove the goodness of the simulations against observations, as they indicate only that the model reproduces the temporal variability observed, but the presence of biases cannot be detected by correlations. In any case, the fact that in some seasons (months? Please see next comment) you have low correlations

points out that there could be differences also in the simulated temporal patterns, at least in some seasons, and this needs to be better discussed. Revise.

Yes, we have revised.
[Lines 177-184]: "*Although the TOMCAT overestimates the SD (**Figure 2a**) because of its biases (Li et al., 2022), it can be seen from **Figure 2a** that TOMCAT matches well the SD variability and correlation coefficients with ENSO in the C3S dataset. These biases of TOMCAT simulation are likely due to (1) the incomplete presentation of complex atmospheric process in the TOMCAT, or (2) the uncertainties in the TOMCAT's meteorology (ERA5) reanalysis scheme (Mitchell et al., 2020; Dhomse et al., 2021). Nevertheless, the high correlation (above 0.95, **Figure 1**) of TP TCO between C3S dataset and TOMCAT simulation from December to May give us confidence that the TOMCAT is able to capture the observed variability in TP TCO during these seasons and that we can thus use it to investigate the impact of ENSO on the TP TCO change*".

26. Figure 1: If the correlations are 3-months, and we have 4 seasons, why do you have 12 columns in the plot?

Sorry for the typo. It is the correlation coefficients between the time series of 3-month running mean TCO anomalies over the TP. We have corrected the caption of Figure 1.

27. Line174: The seasons are already utilized previously, so must be explained previously.

We have added explanation previously [Lines 159-160].

28. Line 177 and also 136-137: The use and the explanation of the lead-lag correlation coefficient is not clear.

We have added its use and explanation into Section 2.4 as noted [Lines 131-133]: "*In order to find out during which months there is a significant response of TP TCO to ENSO, we use lead–lag correlation coefficient, which is calculated according to cross correlation function (Chatfield, 1982).*"

29. Line 183-186: Not clear, revise.

Revised.
[Lines 202-205]: "*Therefore, the following composite TCO anomalies are averaged during December–May to maximize the signal, while we should note the general relationship between ENSO and TCO is not sensitive to the chosen period as one could expect from the positive correlation during December–May*".

30. Line 191: Is it standard deviation or variance?

It is standard deviation. We have revised Figure 2 and associated description [Lines 210-211].

31. Table 1 and 2: What is the meaning of the mean Niño 3.4 index?

It is the mean Niño 3.4 index for total events of El Niño (Table 1) and La Niña (Table 2). We have added it into revised manuscript [Line 220].

32. Lines 211-221: The spatial pattern of the variability is different in the two phases, and should be discussed. I would also recommend discussing the relevance of the changes in percentages (are 8 DU anomalies relevant?) The discussion is in any case not clear and should be revised.

Thank you for this. Now we have added the discussion:
[Lines 223-226]: "The TCO spatial patterns between El Niño and La Niña events are generally opposite despite some differences (Figure 3), which may be related to the asymmetric features in ENSO itself and its climate impacts (Hoerling et al., 1997; An and Jin 2004; Gao et al., 2019)".
We have also added the definition of the relative percentage change.
[Lines 228-229]: "On average, the ENSO events correspond to about ±1.2% relative percentage change (i.e., the anomaly divided by its climate mean) of TP TCO in terms of C3S dataset".

33. Figure 4 and 5: The plot documents that there are significant disagreements between the model and the observations, especially in the La Nina phase where the simulated vertical profile is remarkably different than the observed one. I am not sure if this depends from one particular event which is not simulated correctly or it is a general problem, since the authors have always used the composite seasonal means considering together DJF and MAM rather than analysing single events..

These disagreements are most likely due to the biases of TOMCAT, which may be related to (1) the incomplete presentation of complex atmospheric process in the TOMCAT, or (2) the uncertainties in the TOMCAT's meteorology (ERA5) reanalysis scheme (Mitchell et al., 2020; Dhomse et al., 2021). Please see [Lines 177-184 or Specific comments 25]
To Figures 4-5, we also added the associated discussion as noted [Lines 254-257]: "……*there are disagreements between the TOMCAT and SWOOSH dataset due to the model's biases (Dhomse et al., 2021)……*".

34. Figure 6: Are the spatial changes in TH similar to those in TOC? The spatial pattern seems different, so it is not clear how the two changes can be actually connected. Also, the implied

changes in ozone documented in Figure 6c and d are remarakbly lower than those presented previously...

The difference in their spatial pattern is because the TCO represents total column ozone, but there is a great coherence between the TH and partial column ozone. We have added a plot and discussion [see Figure 6; Lines 280–283]. Please see the detailed response mentioned above (*1ˢᵗ General comment*).

For the second part of the comment, Figures 6 show composite partial column ozone anomalies rather than the TCO, therefore their values are lower than those of Figure 3.

**35. Lines 282-283: Please explain better how TH and the SSTA are linked to atmospheric circulation, and if they are linked. Up to now you were talking of just TH...**

We have revised it as noted: [Lines 301-304]: "*According to equation (3), the anomalous tropospheric upper–level geopotential height can induce the tropospheric temperature change via modifying the air thickness (e.g. Wallace et al., 1996; Sun et al., 2017; Li et al., 2022). As the TH is closely related to tropospheric temperature change (e.g. Seidel and Randel, 2006), it is suggested that the anomalous upper–level geopotential height could influence the TH change.*".

**36. Lines 288-308: the discussion is confused and not straightforward.**

Revised.

[Lines 323-326]: "…… *The possible mechanism for the ENSO teleconnection over the TP has been discussed by previous studies, including excited Rossby wave from tropical Pacific and Indian Ocean to extratropical regions (e.g. Jin and Hoskins, 1995; Trenberth et al., 1998; Zhang et al., 2015b), and enhanced land-sea temperature contrast between tropical Indian Ocean and TP (e.g. Chen and You, 2017; Zhao et al., 2018)……*".

**37. Lines 317-319 and 336-349: The analysis of correlation coefficients alone cannot justify the physical mechanism that you are implying. Please better discuss.**

Yes, we agree that the correlation analysis has its limitation, which we have added a caveat in summary and discussion [Lines 403–407]. In addition, the analysis is shown not only through correlation coefficients but also through physical linkage, including the equation (3) and WMO's TH definition.

[Lines 406–410]: "*Our study focuses on the diagnosed ozone changes over the TP during ENSO episodes using both observations and a chemistry transport model TOMCAT as well as several statistical methods, which will have some uncertainties due to large internal variability of ozone and limited ENSO events. Future work is needed for a better understanding of tangible ENSO impacts with more observed ENSO events and a full-chemistry climate model*".

**38. Figure 8: Yet another spatial pattern, different than those presented previously...**

We have plotted the composite anomalies of the partial column ozone at 150 hPa (Figures 6c–6d), which has a good coherence with the TH (Figure 6a–6b) and tropospheric mean temperature (Figure 8). They all show that the composited anomalies decrease from south to north of the TP.

**39. Figure 9 and line 338: It is not clear the meaning of "temperature associated with air thickness..**

The "temperature associated with air thickness" is calculated from equation (3), indicating the air temperature caused by air thickness. We have added it into caption of Figure 9 [Line 368]

***Technical comments***

**1. Line 19: Perhaps remove "of ENSO"?**

Deleted.

**2. Line 25 and 363: "lead"?**

Yes, it is "lead".

**3. Lines 69-70: I would suggest rephrasing: "The effects of ENSO on ozone changes at mid-latitude and in particular over the TP are less studied and discussed."**

Rephrased.

**4. Lines 70-71: Rephrase "... suggest the amplitude of ENSO signal in TCO over the TP to be of the order of 20 DU (their figure 3)"**

Rephrased.

**5. Line 78: Add "following" after "three"**

Added.

6. Line 86: what do you mean by "merged"?

The "merged" means that this data is created by combining ozone data from 15 satellite sensors. The modification has been made in Section 2.1 of revised manuscript [Lines 82-87].

7. Line 220: Change "understanding" to "understand"

Done.

[revised manuscript text omitted]

---

## Author Comment (AC2)

**Response to the Referee #2's comments on "The impact of El Niño–Southern Oscillation on the total column ozone over the Tibetan Plateau" (egusphere-2023-1452)**

We thank the referee for the helpful comments. His/her insights have improved the quality of our paper. The manuscript has been revised and improved in response to the referee's comments and suggestions. Below is a point-by-point response (in black) to the referee's comments (in blue) followed by any modifications to the manuscript (*in italics*). We have updated explanation for clarification and included a caveat in discussion on the uncertainties of the results.

The line numbers for the changes correspond to the clean/revised manuscript version.

Response to *Anonymous Referee #2*

*General comments*

The article "The impact of El Niño–Southern Oscillation on the total column ozone over the Tibetan Plateau" submitted by Yang Li et al. studies the connection of the ENSO to the total column ozone (TCO) above the Tibetan Plateau (TP). By investigating long-term satellite data from the C3S, the chemical transport model TOMCAT, and the water vapor and ozone data set SWOOSH, the authors connect the positive (negative) anomalies in the Niño 3.4 index to anomalies in the TCO and ozone profiles. The study of this topic is very interesting and the article would be well suited for ACP and an important contribution to the community. In addition, the article is well-written and mostly understandable.

We thank the referee for these positive comments.

There are, however, some aspects to the scientific presentation and content that need major revision before recommendation for publishing. First of all, the analysis restricts to the assessment of anomalies averaged over multiple Niño events, there is no mention of the spread between events. A comprehensive study would benefit greatly from assessing, or even briefly showing, the variability between different events and the dependence of the anomalies on the Niño 3.4 Index.

Thank you. We agree that there would be some spread between events. However, as we have stated in [Lines 355-356] and Figure 9, the relationship is significant with limited spread (p < 0.01), meaning the changes of ozone during the majority of ENSO events are in coherent with the composited anomalies.

Furthermore, the analysis of this article is restricted to correlations between different anomalies. Drawing conclusions on the causation of the TCO stays, therefore, difficult. Especially, since, as the authors mention, the TCO is influenced by multiple effects. There is, however, no apparent attempt to decouple the considered effect of the Niño from the other processes.

Good point. The QBO is another potential important source of interannual variability, which we have now analyzed by adding a plot comparing our results with and without QBO. By doing this, we show that our results are not sensitive to the QBO, and make sure the impact of ENSO on the TP TCO is robust during December–May with or without the QBO signal. Please see [Lines 186–205] and Figure 2.

Yes, we need to be cautious when making conclusions by the statistical methods. We have now added a caveat in the discussion to show the uncertainties and limitation of this study.

[Lines 406–410]: "*Our study focuses on the diagnosed ozone changes over the TP during ENSO episodes using both observations and a chemistry transport model TOMCAT as well as several statistical methods, which will have some uncertainties due to large internal variability of ozone and limited ENSO events. Future work is needed for a better understanding of tangible ENSO impacts with more observed ENSO events and a full-chemistry climate model*".

Lastly, the explanation of the positive TCO anomaly by a downward shift of the ozone profile is lacking. How would a mere downward shift alter the total ozone in a column? Or is partial column ozone considered? The authors should explain the mechanism behind profile shifting leading to increased ozone in a clearer way in order to make it comprehensible.

The shift of ozone profile represents the partial column ozone anomalies associated with TH change. We have updated explanation for clarification.

[Abstract, Lines 28-30]: "*This reduced temperature associated with El Niño events causes a decrease of the tropopause height, which tends to replace ozone–poor tropospheric air by ozone–rich stratospheric air in the UTLS and hence leads to the increase in TCO*".

[Results, Lines 272-276]: "*Approximately 90% of ozone in the atmospheric column resides in stratosphere; the ozone concentration is much lower in the troposphere with a gradual transition at the tropopause. Therefore, a decrease of tropopause height (TH) will tend to replace ozone–poor tropospheric air by ozone–rich stratospheric air in the UTLS region, and thus increase the partial column ozone, which in turn contributes to the TCO increase, and vice versa for an increase of TH (e.g. Schubert and Munteanu, 1988; Salby and Callaghan, 1993; Steinbrecht et al., 1998; Chipperfield et al., 2003; Varotsos et al., 2004; Tian et al., 2007)*".

*Specific comments*

1. Line 47: "high atmospheric transparency" This region is below the Asian Tropopause Aerosol Layer, which should affect the atmospheric transparency as well. Consider adding a comment on its effect.

We have added the associated citation into the revised manuscript [Line 41].

2. Lines 54–58: Please expand a bit on these processes: Brief description of the mechanism (at least for the dominant effects). Is there seasonal varying importance of the different processes?

Yes, the brief description of the mechanism and seasonal change has been revised as noted [Lines 48-57]: "*These studies have argued that the summertime TCO low is caused by changes in mass exchange between troposphere and stratosphere due to the stratospheric variability, for example the synchronisation of the quasi–biennial oscillation (QBO) and seasonal cycle (Chang et al., 2022), and tropospheric changes, for example the high topography and thermal forcing of the TP (Ye and Xu, 2003; Kiss et al., 2007; Tian et al., 2008; Guo et al., 2012) and enhanced convective activity in summer (Liu et al., 2003; Bian et al., 2011). In comparison to the summertime TCO change, less attention has been paid to the TP TCO variability during other seasons. It is worth highlighting that the interannual variability of TP TCO is strongest from wintertime to springtime (Figure S1 of the Supplement). The QBO, a significant natural mode of interannual variability (e.g. Fusco and Salby, 1999; Kiss et al., 2007), could not only contribute to the summertime TP TCO change via modifying the SAH (Chang et al., 2022), but also correlate with wintertime TCO variation (Zhang et al., 2014; Li et al., 2020)*".

3. Lines 62–64: You explain the EN part of ENSO here. Please add a sentence on the Southern Oscillation, i.e., the atmospheric anomalies of the ENSO.

Thanks, done as
[Line 59-61]: "*ENSO represents a periodic fluctuation of the tropical Pacific sea surface temperature (SST) and sea level pressure during warmer phase (El Niño) and colder phase (La Niña)*".

4. Lines 64–69: Why are these regions "showing the significant interannual variability"? Please rephrase or expand.

Rephrased in [Lines 63-66].

5. Lines 72–73: Expand on what limits the satellite measurements. Probably, there is only a limited number of ENSO events in this time period. The sentence could also be understood that there are deficiencies in the measurements themselves. Please clarify.

Clarified.
[Lines 68-69]: "*……their results are based on very limited ENSO events since the satellite era……*".

6. Lines 95–96: "The long-term stability of the TCO product is within the 1% per decade level" It's not clear to me what this means, please expand.

Expanded.
[Line 87-88]: "*The long-term stability of the TCO product with reference to the ground–based monitoring networks is within the 1% per decade level*".

7. Line 103: "and has 12 levels per decade in pressure ranging from 316 to 1 hPa (31 pressure levels)" Do you mean 12 time steps per decade, i.e., 10-monthly data? Please clarify what the "12 levels per decade" refer to.

Sorry for the confusion. Corrected.
[Line 98]: "……*has 31 pressure levels from 316 to 1 hPa*".

8. Lines 109–111: Why not use the SST from ERA5? Please briefly comment.

The ERA5 data is reanalysis data product, which assimilates the observations in the ECMWF model. The HadISST1 only contains observations and has been widely used by groups worldwide.

9. Lines 111–113: Why use the Niño 3.4 index instead of other indices? Please briefly comment.

The Niño 3.4 index has an advantage over other indices for the beginning, end, duration, and magnitude of ENSO events (Trenberth, 1997).

10. Line 158: This is a running 3-month mean, right? Consider stating this in the text.

Yes, we have revised it in [Lines 155-156] and caption of Figure 1 [Line 166].

11. Lines 179–181: Please state that this refers to the bars in Fig. 2.

Done.
[Line 176]: "……*is greater pre–May than post–May (bars in Figure 2a)* ……".

12. Line 185: "as one could be expected from", remove either "one" or change "be expected" to "expect".

Corrected. We have changed "be expected" to "expect" as noted [Line 203].

13. Lines 181–186: The variance is much higher in TOMCAT than in the observations (50–100%). Is this accounted for in the following considerations? I would not call this "reasonable magnitude" but still the variability is well-matched. Maybe you could use the systematic difference found here to put the later results into perspective.

Good point. We have used the systematic difference to explain the possible reason for the differences in later composited results.
[Lines 177-184]: "*Although the TOMCAT overestimates the SD (**Figure 2a**) because of its biases (Li et al., 2022), it can be seen from **Figure 2a** that TOMCAT matches well the SD variability and correlation coefficients with ENSO in the C3S dataset. These biases of TOMCAT simulation are likely due to (1) the incomplete presentation of complex atmospheric process in the TOMCAT, or (2) the uncertainties in the TOMCAT's meteorology (ERA5) reanalysis scheme (Mitchell et al., 2020; Dhomse et al., 2021). Nevertheless, the high correlation (above 0.95, **Figure 1**) of TP TCO between C3S dataset and TOMCAT simulation from December to May give us confidence that the TOMCAT is able to capture the observed variability in TP TCO during these seasons and that we can thus use it to investigate the impact of ENSO on the TP TCO change*".

14. Lines 212–221: Here, all Niño events have been composited. It is unclear how the composition was performed: e.g. average or weighted average according to the Niño 3.4 index? Please specify. In addition, it is unclear what the behavior for individual Niño events is. Please give at least a comment about the variability throughout the different events.

Sorry for this. The composition is calculated by the average of the variable during ENSO events. This has been specified now in Method [Lines 134-135]: "*It is calculated by the average of the variable during ENSO events and its statistical significance is tested by the two–tailed Student's t–test*".
For the second part of the comment, the behavior for individual El Niño event has been shown in the scatter plot with each dot representing a single ENSO event (Figure 9), which has confirmed the significance of the composited results and the relationship between ENSO and ozone-related changes.

15. Figure 4: Plotting the (standard) deviation of the profiles, i.e., the profile ± variability, would be an easy way to show the variability between different events. Consider adding these intervals (e.g. as shading) to the figure.

Thanks for the suggestion. We have revised Figure 4 to show the standard deviation of the profiles.

16. Lines 253–254: It is unclear to me how a downward shift of a profile could singularly alter the total content of ozone in the respective column. Either the partial column ozone is changed, e.g., ozone up to 50hPa, or there has to be an increase in production/mixing from adjacent regions. A stretching (or compression) of the profile, for example, would change the TCO. Please clarify the mechanism that, in the end, leads to increased TCO.

The shift of ozone profile represents the partial column ozone anomalies associated with TH change. We have expanded the explanation to clarify the mechanism. Details, please refer to our response to the **General comment 3**.

17. Line 262: Are the latitude-height sections averaged in longitude or taken as a cross-section at a fixed longitude? Either way, please specify.

Specified by adding "*averaged in longitude (from 75.5°E to 105.5°E)*" in [Line 284].

18. Figure 6: The change in TH alone (located mostly between 30°N –35°N) does not explain the widespread change in ozone stretching to at least 42°N. Locally, I agree that the TH might contribute but do you have hypotheses on the cause of the northern part of the anomalies?

The changes in the northern part of the anomalies are very likely due to the horizontal mixing by advection, in line with previous studies (Neu and Plumb, 1999; Plumb, 2002).

19. Lines 299–304: Could there be a surface temperature anomaly above the TP due to Niño events? If not or of the opposite sign to the Indian Basin SSTA, it could strengthen the argument of the land-sea contrast.

Yes, the 2-m surface air temperature anomaly above the TP is negative during El Niño events, which is of opposite sign to the SSTA in the Indian Ocean. We have added a sentence in our revised manuscript to strength the argument of the land-sea contrast [Lines 327-328].

20. Line 338 & Fig. 9: Specify what "temperature associated with air thickness" refers to more clearly. I suppose this is the temperature as calculated from Eq. 3?

Yes, it is from Eq.3, which has been added it into caption of Figure 9 and [Lines 331-333].

21. Lines 344–346: There are some severe outliers in Fig. 9, e.g., La Nina with -2T thickness. Are the outliers generally corresponding to a weaker Niño index? Consider coloring the scatter plot with the Niño 3.4 index instead of blue/orange. But, of course, there could be various other processes involved in singular events.

Fair point. We have revised Figure 9b using different colours according to the Niño 3.4 index.

*Technical comments*

1. Line 119: "to the 1984–2021." There seems to be a word missing here: average, period?

Thanks. Added.

2. e.g. line 219: "from the December of the ENSO's mature phase to the May of" the use of "the" in front of a month is usually incorrect and reads cumbersomely. Consider removing "the". The same is true for time periods throughout the text, e.g., "the YEAR–YEAR" →"YEAR–YEAR" (unless using a trailing noun such as "the YEAR–YEAR period").

Removed.

3. Line 266: "further results" → "further contribute"

Done.

4. Line 288: Consider dropping "as" in "is considered as an important".

Done.

5. Line 295: "is a response of SSTA" → "is a response to SSTA"

Done.

6. Figure 7: Consider enlarging the text on the color bars.

Done.

**References**

Chipperfield, M. P., Randel, W. J., Bodeker, G. E., Dameris, M., Fioletov, V. E., Friedl, R. R., Harris, N. R. P., Logan, J. A., McPeters, R. D., Muthama, N. J., Peter, T., Shepherd, T. G., Shine, K. P., Solomon, S.,

Thomason, L. W., and Zawodny, J. M.: Global Ozone: Past and Present, in: WMO (World Meteorological Organization) Scientific Assessment of Ozone Depletion: 2002, Global Ozone Research and Monitoring Project – Report No. 47, WMO, Geneva, 498 pp., https://csl.noaa.gov/assessments/ozone/2002/chapters/chapter4.pdf (last access: 27 September 2023), 2003.

Neu, J. L. and Plumb, R. A.: Age of air in a "leaky pipe" model of stratospheric transport, J. Geophys. Res., 104, 19243–19255, https://doi.org/10.1029/1999JD900251, 1999.

Plumb, R. A.: Stratospheric transport, J. Meteorol. Soc. Jpn. Ser. II, 80, 793–809, https://doi.org/10.2151/jmsj.80.793, 2002.

Salby, M. L. and Callaghan, P. F.: Fluctuations of total ozone and their relationship to stratospheric air motions, J. Geophys. Res., 98, 2715-2727, https://doi.org/10.1029/92JD01814, 1993.

Schubert, S. D. and Munteanu, M. J.: An analysis of tropopause pressure and total ozone correlations, Mon. Weather Rev., 116, 569-582, https://doi.org/10.1175/1520-0493(1988)116<0569:AAOTPA>2.0.CO;2, 1988.

Steinbrecht, W., Claude, H., Köhler, U., and Hoinka, K. P.: Correlations between tropopause height and total ozone: Implications for long-term changes, J. Geophys. Res., 103, 19183-19192, https://doi.org/10.1029/98JD01929, 1998.

Tian, B., Yung, Y. L., Waliser, D. E., Tyranowski, T., Kuai, L., Fetzer, E. J., and Irion, F. W.: Intraseasonal variations of the tropical total ozone and their connection to the Madden-Julian Oscillation, Geophys. Res. Lett., 34, https://doi.org/10.1029/2007GL029451, 2007.

Tian, W., Chipperfield, M., and Huang, Q.: Effects of the Tibetan Plateau on total column ozone distribution, Tellus B, 60, 622-635, https://doi.org/10.1111/j.1600-0889.2008.00338.x, 2008.

Trenberth, K. E.: The Definition of El Niño. Bull. Amer. Meteor. Soc., 78, 2771–2778, https://doi.org/10.1175/1520-0477(1997)078<2771:TDOENO>2.0.CO;2, 1997.

Varotsos, C., Cartalis, C., Vlamakis, A., Tzanis, C., and Keramitsoglou, I.: The long-term coupling between column ozone and tropopause properties, J. Clim., 17, 3843-3854, https://doi.org/10.1175/1520-0442(2004)017<3843:TLCBCO>2.0.CO;2, 2004.

Zhang, J., Tian, W., Xie, F., Tian, H., Luo, J., Zhang, J., Liu, W., and Dhomse, S.: Climate warming and decreasing total column ozone over the Tibetan Plateau during winter and spring, Tellus B, 66, 23415, https://doi.org/10.3402/tellusb.v66.23415, 2014.

---

## Referee Report (RR1)

**Review of egusphere-2023-1452 "The impact of El Niño–Southern Oscillation on the total column ozone over the Tibetan Plateau" by Yang Li et al.**

December 13, 2023

**General comments**

The manuscript of "The impact of El Niño–Southern Oscillation on the total column ozone over the Tibetan Plateau" by Yang Li et al. focuses on the effect of the ENSO on total column ozone (TCO) above the Tibetan Plateau (TP). Through analysis of the aggregated long-term satellite data C3S, the chemical transport model TOMCAT, and the water vapor and ozone dataset SWOOSH, the authors present statistical significant relations of positive TCO anomalies above the TP during El Niño and vice versa for La Niña events. These anomalies are robustly found in all presented datasets.

This study is definitely relevant and suits the scope of ACP well and is a welcome addition for the atmospheric community. The manuscript itself is mostly well written and understandable. However, there are parts, that are less well described and early conclusions that cannot be followed in the current state of the article. Hence, a **major revision** of this article are needed before a recommendation for publishing can be made.

In particular, the authors only look at correlations of different parameters and jump to the conclusion of a causation. The whole mechanism of the TCO increase described in the conclusion is not well supported by the findings of the study and could be argued for in different order as well. The ENSO is leading the TCO anomaly but it is not clear if the cooling of the troposphere or the lowered geopotential height of the 150 hPa level is the next step in the mechanism. Since only the correlations are shown, this conclusion cannot be drawn.

Furthermore, the increase of the TCO is argued by the reduction in tropospheric height and a subsequent "change in the relative amounts of ozone–poor tropospheric and ozone–rich stratospheric air in the profile". While this is true for the relative amounts of air, it is not necessarily for the TCO. The TCO can only increase if there is an influx of stratospheric, ozone-rich air from other regions. This is not talked about although it is the center point of this study.

In general, the authors should either rephrase their claims or increase their evidence for the presented mechanism. In the current state, the stated causation is not evident.

Following are specific and technical comments with citations from the manuscript given in italics.

**Specific comments**

- l 57 ff: There is no mention abut the atmospheric part of the ENSO, i.e., the SO part. Add a sentence describing this as well.

- l 99: how many vertical levels does your data span between 1000 and 0.01 hPa?

- l 122: *which is calculated according to cross correlation function (Chatfield, 1982)*: probably an article missing (a or the?), but the cross correlation function should be described by a bit more detail. Or the sentence should be rephrased for better understandability.

- l 224: *±1.2% percentage change (i.e., anomaly divided by climate mean) of TP TCO* You stated that your dataset has an accuracy of about 2%, which would render this change to be within the expected error if found in an individual case. Please comment on the this discrepancy within the text. Since you use composite events, the errors are canceling to some extent. Do you have a feeling to what extent?

- l 225 ff: *To further clarify the influence of ENSO on TP TCO, we use regression method to remove the ENSO signal from TP TCO and then perform composite TCO during El Niño and La Niña years. Without considering the ENSO signal on the TP TCO, both C3S data and TOMCAT simulation show that there is no significant TCO anomalies over the whole TP during El Niño and La Niña years (not shown). As it is, this section gives no new information since it basically reads as: if we remove the linear relation,*

we find no longer a linear relation. Please describe how you removed the ENSO signal. This should be a physically justified approach for the separation. If it is actually a regression separation, remove this part.

- Fig. 3: There is a shaded region mentioned for the stat. significance, but not shown. Change this to state something like: 'There is no region, where the anomaly is statistically significant beyond the 90% confidence level.' in the caption.

- l 243 ff: although the altitudes of 100–50 hPa agree with your vertical shift hypothesis, the Ozone profiles between 200–100 hPa show a different slope for El Niño and La Niña. This might not only be related to a vertical shift but a compression/stretching of the profile for Niño and La Niña, respectively?

- l 270 ff: *Therefore, a decrease of tropopause height (TH) will tend to replace ozone–poor tropospheric air by ozone–rich stratospheric air in the UTLS region ...* This would only be true, if there is horizontal flow of from surrounding areas towards the TP. In other words, a constant Stratopause would be expected and/or an opposite TCO anomaly adjacent to the TP. The Ozone has to come from somewhere. Please add on this question for a more detailed description of this process. As it is right now, your point is not self-sufficient.

- l 331 f: *That is, the tropospheric temperature will warm (cool) when the rising (falling) GH150 causes the increased (decreased) air column thickness.* This sounds too much like a cause and consequence where only correlation is seen. Please rephrase this sentence accordingly.

- l 339 f: *According to equation (3), this implies that the El Niño (La Niña) events favour the decreased (increased) air thickness and thus contributes to the cooling (warming) tropospheric mean temperature (Figures 8).* 'Contributes' is more than you can claim here, I would say it is 'associated with' since you only found the correlation.

- l 351 f: *... there is a strong negative correlation (–0.56) between Niño 3.4 and temperature associated with air thickness.* While it is very convincing that the air thickness is related to whether there is a El Niño or La Niña, the correlation to Niño 3.4 is not as convincing. Did you have a look at other indices and their correlation to the air thickness? And what about the non-ENSO events in the same periods, do they fall close to the center point in this diagram, or are they scattered on the vertical axis? Please comment on this in the text.

- l 378 f: *...with and without ENSO signal...* should be '...with and without QBO signal...' if I'm not mistaken.

- l 393 ff: *Thirdly, such a TH decrease tends to cause a change in the relative amounts of ozone–poor tropospheric and ozone–rich stratospheric air in the profile, which increases the partial column ozone in the UTLS...* See above. The partial column ozone will change for a fixed pressure interval, but for the TCO, there needs to be an influx (or source) of Ozone adjacent to the column. The process described here is not clear. The main question I'm left with is: where is the additional Ozone in the TCO stemming from?

**Technical comments**

- l 67: since the satellite era → during the satellite era

- l 104: of area averaged of SST anomalies (SSTA) → of area averaged SST anomalies (SSTA)

- l 104 f: consider adding a reference to the website here

- l 106 f: *As the SWOOSH spans from 1984 to the present, its anomalies are with respect to the period 1984–2021.* consider rephrasing this for better understanding

- l 131 f: *In view of the fact that the relationship between the positive and negative ENSO phases may not be linear (An and Jin 2004), we consider El Niño and La Niña events should be analyzed separately.* remove 'should be analyzed' or 'we consider'

- l 187: presentation → representation

- sec 3.1: use *lead-lagged* (or *lagged-lead* if you prefer this) consistently in your text and captions

- l 247: remove *Such*

- l 250: remove *, whose results are consistent with Figures 5a and 5d.* Since its only a zoom it is not only consistent but identical. No need to clarify after stating it's a zoom.

---

## Referee Report (RR2)

**Review of egusphere-2023-1452 "The impact of El Niño–Southern Oscillation on the total column ozone over the Tibetan Plateau" by Yang Li et al.**

February 26, 2024

**General comments**

The article "The impact of El Niño–Southern Oscillation on the total column ozone over the Tibetan Plateau" by Yang Li et al. investigates anomalies of total column ozone over the Tibetan Plateau During El Niño and La Niña events. As for the newest revision, I recommend this article for publication after some minor changes (mostly technical).

Most issues have been resolved by the changes and the author's comments. In particular, the revised version has clarified the methodology and the text throughout the article. Furthermore, the revised text reflects the data and derivable results much better than before.

Following are some minor comments that should be addressed before acceptance to ACP.

**Minor comments**

- Consider adding the discussion of Table A1/B1 and Figure A1/B1 to the Appendix (or at least to the supplement) since it supports your analysis and findings.

- l 198: "QBO signal ($TCO_{removeQBO}$) can be written as follows:" is there a reference for the linear separation of the QBO effect? If so, please add it here.

- Fig. 6 caption: The difference between "thick green lines" and "green lines" is barely visible. I suggest increasing the width of the thick line. Also consider shrinking the vertical axis in order to enhance visibility of the "shaded area" around the green lines.

**Technical comments**

- l 195: ".. another potential important source of interannual variability." -¿ "another potential important source of interannual variability of TCO above the TP."

- l 335 "...thickness and thus associated with...": There si a word missing. E.g., either "are thus associated" or "thus are associated".

- l 379 "...to MAM in decaying phase...": Either "in **its** decaying phase" or "in **the** decaying phase"

---

## Author Response (AR2)

**Response to the Referee #1's comments on "The impact of El Niño–Southern Oscillation on the total column ozone over the Tibetan Plateau" (egusphere-2023-1452)**

We again thank Referee #1 for making detailed comments and very useful suggestions to improve the paper. The manuscript has been revised and further improved in response to the referee's comments and suggestions. Below is a point-by-point response (in black) to the comments (in blue) followed by the corresponding changes in the manuscript with track changes (*in italics*). Some key general points relevant to the comments are:

- In the first revised version, we used correlation coefficient, composite method, hypsometric equation, and TOMCAT results to study our topic, and we had a caveat of correlation analysis in summary and discussion.
- In the first revised version, the high correlation between C3S data and TOMCAT (from DJF to MAM) gives us confidence that the TOMCAT is able to capture the observed variability in the total column ozone (TCO) over Tibetan Plateau (TP), and we also discussed the reasons of TOMCAT biases.
- In the new revision, we have now also compared the TP TCO with and without considering the ENSO signal to highlight the influence of ENSO on TP TCO.
- In the new revision, we have now added more details on Student's *t*-test and its statistical significance, and we also have revised the statistical significance of spread.
- In the new revision, we have further improved the abstract and conclusion.
- In the new revision, we have modified a caveat in the discussion to show the limitation and indication of Figure 9.
- In the new revision, the Tables 1-2 have been moved to method section and the download links of datasets have been moved to data availability.

The line numbers for the changes correspond to the marked-up manuscript version.

Response to *Anonymous Referee #1*

*General comments*

With respect to the original version of the manuscript, I believe that this revised version has addressed some of the main concerns raised by both reviewers during the first revision. However, many doubts remain and are not totally solved with the author's responses and changes. In particular, doubts remain about the almost exclusive use of the correlation coefficients, and the use of the Student's t-test to infer about its statistical significance (which is not clear) and also on the use of the statistical significance to infer about the spread. Indeed, both reviewers previously raised comments against the use of only anomalies averaged over multiple Niño events, without

an analysis of the spread of the events. Similarly, both reviewers raised comments about the use of only correlation coefficients to draw conclusions on the TCO causal mechanisms. In this respect, I consider the responses and the changes applied not satisfactory enough. I urge the authors to carefully reconsider those comments and to greatly improve the analysis in this regard. Indeed, I believe that, those being the major comments raised by both reviewers during the first revision phase, they deserve better attention by the authors. The rest of more specific comments is provided below.

We thank the reviewer for suggesting that we reconsider the statistical method used in the analysis. In the first revised version, we used correlation coefficient, composite method, hypsometric equation, and TOMCAT results, and we had a caveat of correlation analysis. Now we have added new information included in the revision:

1) composite of TP TCO with and without the ENSO signal
2) more details on Student's *t*-test
3) correction of spread's statistical significance

Many studies have used correlation coefficients to investigate the linkage between two factors (e.g. Thompson and Wallace, 1998; Woolnough et al., 2000; Trenberth et al., 2015). Based on this point, we use correlation coefficient to find the relationship between ENSO and TP TCO. Our results show that there is a robust linkage between ENSO and TP TCO from DJF to MAM (Figure 2). To further make sure that our conclusion is convincing and the result is robust, in addition to the correlation coefficient, we reiterate that we also used the composite method, hypsometric equation, and TOMCAT results to study the linkage between ENSO and TP TCO. Our results show that the El Niño (La Niña) events correspond to the significantly positive (negative) TCO anomalies over the whole TP (Figure 3). To further clarify the influence of ENSO on TP TCO, we have added new information on composite analyses of TP TCO with and without ENSO signal (Figure 3 and Figure A1). Through comparing their difference, we find there is no significant TCO anomalies over the whole TP during El Niño and La Niña years when the ENSO signal is removed (Figure A1). This supports that ENSO has an influence on TP TCO. As correlation follows Student's *t* distribution with effective number, we use Student's *t*-test to infer about its statistical significance (please also see responses to *Specific comments 8 and 10*). However, the Student's *t*-test cannot infer the statistical significance of spread and we have revised the associated sentence (please see more responses to *Specific comment 15*). It is also noted that the correlation analysis has its limitation, which was the reason that we had added a caveat in summary and discussion in the first revised version.

Our detailed explanations and discussion of this point are as follows:

**(1) The purpose of using correlation coefficient and the reason of using Student's *t*-test**. In our study, we aim to find the linkage between ENSO and TP TCO variability by using observations and TOMCAT. For this purpose, we should first evaluate the performance of TOMCAT on TP TCO variability. Considering that correlation coefficient can evaluate whether model reproduces the observed variability (e.g. Wang et al., 2008), we use the correlation coefficient to evaluate the performance of TOMCAT on TP TCO variability. The high correlation coefficients (above 0.95

from DJF to MAM) give us confidence that TOMCAT is able to capture the observed variability in TP TCO. We can thus use it to investigate the impact of ENSO on the TP TCO variability (please also see response to *Specific comment 11*). In addition, many studies have used the correlation coefficient to investigate the linkage between two variables (e.g. Thompson and Wallace, 1998; Woolnough et al., 2000; Trenberth et al., 2015). Based on this point, we use the correlation coefficient to find the linkage between ENSO and TP TCO variability. Since the correlation follows Student's *t* distribution with effective number, the two-tailed *t*-test can provide some indication about the significance of the correlation. Please see more responses of Student's *t*-test and statistical significance to *Specific comments 8 and 10*. The Student's *t*-test cannot be used to evaluate the statistical significance of spread. We have revised the associated sentence, please see more responses to *Specific comment 15*.

**(2) In addition to the correlation coefficient, composite, hypsometric equation, and TOMCAT results are also used in our study.** In view of the fact that the relation between the positive and negative ENSO phases may not be linear, we further consider the two phases separately. For this purpose, we perform composite analyses for both phases to further investigate the linkage between ENSO and TP TCO. The composite method has been used by many previous studies (e.g. Thompson and Wallace, 2001). Figure 3 shows the El Niño events correspond to positive TCO anomalies over the TP, and vice versa for the La Niña events. We also use the hypsometric equation (equations 2-3) to investigate the influence of air thickness on tropospheric temperature. In addition, many studies show that TOMCAT performs well in reproducing the observed ozone variability (e.g. Feng et al., 2005; Singleton et al., 2005; Rösevall et al., 2008; Kuttippurath et al., 2010; Chipperfield et al., 2017, 2018; Griffin et al., 2019; Bognar et al., 2021; Feng et al., 2021; Li et al., 2022), due to information from field campaigns/lab measurements/reanalysis dataset and improved tracer transport and chemistry etc. In our study, we find that ENSO has an influence on TP TCO variability from DJF to MAM. During this period, the correlation coefficients between TOMCAT and C3S are above 0.95, suggesting that TOMCAT reproduces well the observed TCO variability.

**(3) Comparing the composite analyses of TP TCO with and without ENSO signal, it is further suggested that ENSO has an influence on TP TCO.** We use a regression method to remove the ENSO signal from TP TCO and then perform composite TCO during El Niño and La Niña years. This method of removal is similar to previous studies (e.g. Thompson et al., 2008; Li et al., 2017). After removing the ENSO signal, Figure A1 shows that there is no significant TCO anomaly over the whole TP during El Niño and La Niña years for C3S dataset and TOMCAT simulation. Comparing this result (Figure A1) to the composite with ENSO signal (Figure 3), it is suggested that ENSO has an influence on TP TCO. These discussions have been added into the revised manuscript [Lines 250–254]: "*To further clarify the influence of ENSO on TP TCO, we use a regression method to remove the ENSO signal from TP TCO and then perform composite TCO during El Niño and La Niña years. Without considering the ENSO signal on the TP TCO, both C3S data and TOMCAT simulation show that there is no significant TCO anomalies over the whole TP during El Niño and La Niña years (not shown). Comparing the TP TCO with and without the ENSO signal supports that ENSO has an influence on TP TCO*".

**(4) We added a caveat of correlation analysis in summary and discussion of first revision**.
[Lines 440–443]: "*Our study focuses on the diagnosed ozone changes over the TP during ENSO episodes using both observations and a chemistry transport model TOMCAT as well as several statistical methods, which will have some uncertainties due to large internal variability of ozone and limited ENSO events. Future work is needed for a better understanding of ENSO impacts with more observed ENSO events and a full-chemistry climate model*".

[Figure]

**Figure A1: Same as Figure 3, but without considering the ENSO signal on the TP TCO.**

*Specific comments*

1. Lines 35-40: The explanation of the mechanism is not that clear. I would suggest trying to improve it.

OK. We have improved it.
[Lines 28–34]: "*Our results suggest that the El Niño events impact the TP TCO via the following processes: (1) negative upper–level geopotential height anomaly associated with El Niño is responsible for a decrease in air column thickness; (2) the thickness decrease modulates reduced tropospheric temperature and thus favours a decrease in the tropopause height (TH); (3) such a TH decrease tends to cause a change in the relative amounts of ozone–poor tropospheric and ozone–rich stratospheric air in the profile, which increases the partial column ozone in the UTLS and hence contributes to the TP TCO increase*".

2. Lines 38-39: How a "decrease in the tropopause height" tends to "replace ozone-poor tropospheric air by ozone-rich stratospheric air in the UTLS"? Not clear. Revise.

The fact is that approximately 90% of ozone in the atmospheric column resides in stratosphere; the ozone concentration is much lower in the troposphere with a gradual transition at the

tropopause. Our Figures 6e-6f show that the TH decrease corresponds to the positive partial column ozone anomaly in the UTLS region, and vice versa for the TH increase. The TH decreases suggest that ozone–poor tropospheric air will be replaced in the profile by ozone–rich stratospheric air, which is supported by many studies (e.g. Schubert and Munteanu, 1988; Salby and Callaghan, 1993; Steinbrecht et al., 1998; Chipperfield et al., 2003; Varotsos et al., 2004; Tian et al., 2007). We have revised in [Lines 31-34]: "……*such a TH decrease tends to cause a change in the relative amounts of ozone–poor tropospheric and ozone–rich stratospheric air in the profile, which increases the partial column ozone in the UTLS and hence contributes to the TP TCO increase*".

3. Lines 42-45: But if the two phases have effects of opposite signs, and both phases will increase in frequency under a global-warming scenario (also, which one?), wouldn't be the final change almost null? Explain better.

We have deleted it and revised our implications as shown in [Lines 35-37]: "*This work provides a systematic understanding of the influence of ENSO on ozone over the TP, which have implications for a better understanding of factors controlling the interannual variability of ozone*".

4. Lines 51-53: How is the "clean air" connected with "strong surface solar ultraviolet (UV) radiation"? Explain better.

We apologise for this unclear sentence. We mean air that has a low concentration of aerosols. According to the Aerosol Robotic Network-based studies (e.g. Pokharel et al., 2019), the clean air corresponds to low aerosol optical depth. Since aerosol optical depth represent how much direct sunlight is prevented from reaching the ground by aerosol particles, the clean air over the TP corresponds to the strong sunlight and surface solar UV radiation (Pokharel et al., 2019). We have revised it in [Lines 45–46]: "*……clean air with low aerosol optical depth (Ahrens and Samson, 2011; Pokharel et al., 2019), the TP experiences strong sunlight and surface solar ultraviolet (UV) radiation……*".

5. Line 82: Still not clear, consider adding the number of events.

Sorry for this unclear sentence. We have added in [Line 73]: "*……limited ENSO events (4 El Niño, 3 La Niña)……*".

6. Lines 151-152: Why do you run simulations for the 1950-2022 period if the overlapping period is only 1979-2021. Please explain.

Good point. The model's initial simulations are unreliable as the model attempts to stabilize, which is the 'spin up' period (e.g. Birner et al., 2020). The selected overlapping period (1979-2021) could

avoid the 'spin up' period of model. We have revised in [Lines 132-134]: "*The overlapping period between model and observations (i.e. 1979–2021) is used for analysis*".

7. Lines 154-155: The identification of the tropopause according to WMO is not a statistical method. Revise.

Sorry for this. We have removed the first sentence of section 2.3 [Line 136]: "".

To make it clear, more details have been added into revised manuscript, shown as follows:

1) [Lines 151–152]: "*……we perform composite analyses of ozone, SST, geopotential height, tropopause height, and air temperature during El Niño and La Niña events……*".

2) [Lines 154–155]: "*The statistical significance of composite is tested by the two–tailed Student's t–test (i.e. Kiladis and Diaz, 1989; von Storch and Zwiers, 1999)*".

3) Based on the WMO's technical note (WMO, 1966) and statistical book (von Storch and Zwiers, 1999), the correlation follows Student's *t* distribution with effective number ($N^{eff}$). Therefore, the two-tailed *t*-test can provide some information about the significance of the correlation. We have added in [Line 146]: "*……where $r_{XY}$ is correlation coefficient between two sampled time series (X and Y), and t value of $r_{XY}$ follows Student's t distribution with $N^{eff}$*".

9. Lines 166-177: Also these methods are not statistical. Revise.

Yes. We have removed the first sentence of section 2.3 [Line 136]. Please see the response to *Specific comment 7*.

10. Lines 183-184: Again, it is not clear how the t-test can be used to infer about the statistical significance of the correlation.

We apologise for still not making this clear. We have now added more details on *t*-test and its statistical significance into revised manuscript. [Lines 141–149]: "*The statistical significance of the correlation between two auto–correlated time series is calculated using the two–tailed Student's t–test and the effective number ($N^{eff}$) of degrees of freedom (WMO, 1966; von Storch and Zwiers, 1999; Pyper and Peterman, 1998; Li et al., 2013), as given by the following approximation:*

$$t = r_{XY} \sqrt{\frac{N^{eff}}{1 - r_{XY}^2}}$$

(1)

$$\frac{1}{N^{eff}} \approx \frac{1}{N} + \frac{2}{N} \sum_{j=1}^{N} \frac{N - j}{N} \rho_{XX}(j) \rho_{YY}(j)$$

where $r_{XY}$ *is correlation coefficient between two sampled time series (X and Y), and t value of* $r_{XY}$ *follows Student's t distribution with* $N^{eff}$; *N is the sample size, and* $\rho_{XX}$ *and* $\rho_{YY}$ *are the autocorrelations of two sampled time series, X and Y, respectively, at time lag j. Based on the two–tailed Student's t–test, the* $r_{XY}$ *can be tested for statistical significance by solving for t in equation (1) and comparing this with the t value at confidence level and* $N^{eff}$*"*.

11. Lines 179-191: The consideration of only the correlation coefficients and their statistical significance (provided it is explained how the statistical significance of the t-test can provide indications on the correlation, which is not the case as noted previously) means that you are evaluating only that TOMCAT and C3S present the same time pattern in the (seasonal) TCO anomalies, while you are not saying anything on the presence of biases. Also, it would be interesting to evaluate the reason why the correlation is significantly lower (higher) in some particular seasons.

Thank you. Our responses to this comment can be listed as follows:

1) Since our results focus on the impact of ENSO on TP TCO variability from DJF to MAM, the high correlation (above 0.95, Figure 1) of TP TCO between TOMCAT and C3S data gives us confidence that the TOMCAT is able to capture the observed variability in TP TCO. We can thus use it to investigate the impact of ENSO on the TP TCO variability.

2) We agree that there are biases between C3S data and TOMCAT results. We have discussed the biases in first revised version as shown in [Lines 206–208 of the new revised manuscript]: "*These biases of TOMCAT simulation are likely due to (1) the incomplete representation of complex atmospheric process in TOMCAT, or (2) the uncertainties in TOMCAT's meteorology (ERA5 reanalysis scheme) (Mitchell et al., 2020; Dhomse et al., 2021)*".

3) **Some discussion about lower correlation.** Figure 1 shows relatively low correlation coefficients between C3S data and TOMCAT simulation from JJA to SON (correlations range from 0.65 to 0.8). We plot the time series of TP TCO anomaly averages for C3S and TOMCAT from JJA to SON (Figure A2). Figure A2 shows that TOMCAT performs reasonably well since 2014, but TOMCAT does not match C3S data well during the periods 1979-1986 and 2008-2013, and for the year 1995. TOMCAT is not a perfect model and its relatively low correlation with C3S is likely due to the convection parameterization. Since the convection parameterization simplifies atmospheric convective processes (e.g. Wu et al., 2011), it may affect the simulated tracer transport results. Given that TP convection is relatively strong from JJA to SON (Zhao et al., 2018),

this may cause more biases in TOMCAT. As the convection is weak from DJF to MAM (Zhao et al., 2018), TOMCAT performs reasonably well and its correlations with C3S are above 0.95.

[Figure]

**Figure A2: Time series of TCO (DU) anomaly averages over the TP for C3S data (blue) and TOMCAT result (red) from JJA to SON.**

12. Lines 217-220: These details are more suitable for other sections.

Good point. We have moved the QBO index to section 2.2 [Lines 118–119]. In addition, its download link has been moved to Data Availability [Lines 452–453].

13. Lines 242-244: Shouldn't these details and methods be presented in the methods section?

Yes, a good point. We have moved these details and Tables 1-2 into the methods section.
[Lines 155–165]: *"……The persistent ENSO events are based on the definition of NOAA Climate Prediction Center and our lead–lagged correlation results. We identify persistent El Niño (La Niña) events if their anomalous Niño 3.4 index is greater (less) than 0.5 K (–0.5 K) from winter (December–January–February; DJF) to spring (March–April–May; MAM). There are 7 and 9 events of persistent El Niño and La Niña (**Tables 1–2**), respectively……".*

14. Lines 320-391: Still not much clear the link between ENSO and change in TH (the link between TH and O3 content is instead straightforward). This improved explanation should be added to the conclusions and also in the abstract.

The improved explanation has been added into the conclusions and abstract in the revised manuscript.

Conclusion [Lines 411–426]: "*Regarding the El Niño events, its linkage with TP TCO is as follows: El Niño → negative upper–level geopotential height anomaly → thickness decrease → reduced tropospheric temperature → TH decrease → TCO increase, where the arrows show the cause-and-effect relationships. It is suggested that El Niño can trigger the TP TCO change via the following processes. Firstly, the El Niño events tend to exert the negative upper–level geopotential height anomaly (**Figure 7**) via the El Niño teleconnection and land-sea temperature contrast associated with El Niño, thus leading to the decreasing air thickness (**Figures 8a–8b**) based on equation (2) and previous studies (e.g. Wallace et al., 1996; Sun et al., 2017). Secondly, according to equation (3), the thickness decrease could reduce the tropospheric temperature over the TP (**Figures 8c–8d**), which further induces a decrease of TH (**Figure 6**) in terms of the tropopause definition of WMO (1957) and our results (**Figure 9**). Thirdly, such a TH decrease tends to cause a change in the relative amounts of ozone–poor tropospheric and ozone–rich stratospheric air in the profile, which increases the partial column ozone in the UTLS (**Figures 4 and 6**) and thus contributes to the TCO increase (**Figure 3**).The linkage between La Niña events and TP TCO as well as its associated processes resembles the El Niño events, except with anomalies of opposite sign*".

Abstract is in [Lines 28–34]. Please see the responses to *Specific comment 1*.

**15. Line 425: The statistical significance does not provide any indication about the presence or absence of the spread.**

Yes, you are correct. The *p* value is used to evaluate the statistical significance of correlation relationship between ENSO and Thickness. We have revised the sentence. [Lines 378–379]: "*The relationship is significant ($p < 0.01$ or above the 99% confidence level), meaning the changes of ozone during the majority of ENSO events are coherent with the composite anomalies (**Figure 8**)*".

**16. Figure 9: 1) the correlation coefficient between change in thickness is not high; 2) there are some events for which the change in thickness is very limited**

Our responses to this comment can be listed as follows:

1) Correlation coefficients are -0.56 and -0.85 in Figure 9. Many studies show that the correlation coefficient is high when its absolute value is above about 0.55. For example, correlation analyses in Trenberth (1975), Thompson and Wallace (1998), Mishra et al. (2011), Trenberth et al. (2015), and Zhang et al. (2014).

2) Yes, we agree. The results of significant correlation coefficient ($p < 0.01$) reveal that there is a robust linkage between ENSO and thickness. The fact is that ENSO can explain a part of total variability of thickness. There may be other factors impacting thickness. We have modified a caveat in the discussion to show the limitation and indication of Figure 9.

[Lines 433–439]: *"Although **Figure 9** is in good agreement with our study and shows that there are significant correlations between samples of ENSO events and air thickness as well as TH, it is also apparent a few samples deviate from the regression line and have the limited change. This implies that in addition to ENSO, there may be other factors contributing to the air thickness and TH variability and thus contributing to the TCO variation. Recently, Duan et al (2023) stated that the tropical Indian Ocean SSTA could cause a vertical shift of the ozone profile over the TP and thus contribute to the TCO variation......".*

**Technical comments**

1. Lines 15-16: change "local ozone in the stratosphere" to "local stratospheric ozone concentration".

Done.

2. Lines 19-20: Change "that there exists a positive correlation.." to "the existence of positive correlation.."

Changed.

3. Line 34: Change "which is the …" to "i.e. in the upper troposphere and lower stratosphere regions."

Done.

4. Line 38: Change "decrease of" to "decrease in".

Changed.

5. Line 49: Change "dominating" to "driving".

Revised.

6. Line 55: Add "concentration variability" after "ozone".

Added.

7. Line 67: Explain what is "SAH".

"South Asian high" is now added.

8. Line 75: Change "on ENSO.." to "about ENSO.."

Changed.

9. Lines 78-79: Change "considering the major ozone production in the tropics…" to "as those two regions correspond to the highest production (tropics) and depletion (poles) on Earth."

Changed.

10. Lines 99-137: the links and detailed information on the access date are not suitable for the main text, but need to be included in the reference (or data availability) sections. In the main text, following a more suitable format for references. Revise.

Revised.

11. Line 124: You mean "tropopause height"?

Yes, we have revised.

12. Line 136: Delete "that".

Deleted.

13. Lines 148-150: Change "and calculates" to "and was used to calculate…"

Changed.

14. Lines 156-157: Change "…to find out during which periods there is a significant response…" to "the months when there is a …"

Changed.

15. Line 183 and 446: Change "strong" to "high".

Changed.

16. Line 198: Change "lag-lead" to "lead-lagged".

Changed.

17. Line 200: This sentence is not clear. Revise.

We apologise for this. This sentence is consistent with caption of Figure 2. We have deleted it.

18. Line 217: Change "then perform …" to "then perform lagged-lead correlation, shown in Figure 2a."

Changed.

19. Lines 236-240: Rephase: "a) Lagged-lead correlation … b) same as a), but without considering the QBO signal on the TP TCO.

Rephrased.

20. Lines 242-244: The sentence is not clear, rephrase.

Rephrased, please see [Lines 155–158]

21. Line 245: Remove "the".

Removed.

22. Lines 247-248: Remove such details about the identification from the caption.

Removed.

23. Line 255-257: Same as noted previously about the links and details about access date.

Revised.

24. Line 268: Change "in terms of" to "as evaluated in the"

Changed.

25. Line 269: Change "composited" to "composite".

Changed.

26. Line 274: Change "to" to "until"

Changed.

27. Line 287: Change "of" to "in"

Changed.

28. Line 426: Delete "in" and change "composited" to "composite".

Deleted and changed.

29. Line 451: Change "lead" to "lagged-lead".

Changed.

**References**

Ahrens, C.D., Samson, P.J.: Extreme Weather and Climate, 1st Edn. Brooks Cole, 508pp, 2011.

Bognar, K., Alwarda, R., Strong, K., Chipperfield, M. P., Dhomse, S. S., Drummond, J. R., Feng, W., Fioletov, V., Goutail, F., Herrera, B., Manney, G. L., McCullough, E. M., Millán, L. F., Pazmino, A., Walker, K. A., Wizenberg, T., and Zhao, X.: Unprecedented spring 2020 ozone depletion in the context of 20 years of measurements at Eureka, Canada, J. Geophys. Res., 126, e2020JD034365, https://doi.org/10.1029/2020JD034365, 2021.

Birner, B., Chipperfield, M. P., Morgan, E. J., Stephens, B. B., Linz, M., Feng, W., Wilson, C., Bent, J. D., Wofsy, S. C., Severinghaus, J., and Keeling, R. F.: Gravitational separation of Ar/N2 and age of air in the lowermost stratosphere in airborne observations and a chemical transport model, Atmos. Chem. Phys., 20, 12391–12408, https://doi.org/10.5194/acp-20-12391-2020, 2020.

Cai, W., Wang, G., Santoso, A. et al: Increased frequency of extreme La Niña events under greenhouse warming. Nature. Clim. Change., 5, 132–137, https://doi.org/10.1038/nclimate2492, 2015.

Cai, W., Wang, G., Dewitte, B. et al: Increased variability of eastern Pacific El Niño under greenhouse warming. Nature, 564, 201–206, https://doi.org/10.1038/s41586-018-0776-9, 2018.

Collins, M., An, SI., Cai, W. et al: The impact of global warming on the tropical Pacific Ocean and El Niño. Nature. Geosci., 3, 391–397, https://doi.org/10.1038/ngeo868, 2010.

Chipperfield, M. P., Randel, W. J., Bodeker, G. E., Dameris, M., Fioletov, V. E., Friedl, R. R., Harris, N. R. P., Logan, J. A., McPeters, R. D., Muthama, N. J., Peter, T., Shepherd, T. G., Shine, K. P., Solomon, S., Thomason, L. W., and Zawodny, J. M.: Global Ozone: Past and Present, in: WMO (World Meteorological Organization) Scientific Assessment of Ozone Depletion: 2002, Global Ozone Research and Monitoring Project – Report No. 47, WMO, Geneva, 498 pp., https://csl.noaa.gov/assessments/ozone/2002/chapters/chapter4.pdf (last access: 27 September 2023), 2003.

Chipperfield, M. P., Bekki, S., Dhomse, S., Harris, N. R. P., Hassler, B., Hossaini, R., Steinbrecht, W., Thiéblemont, R., and Weber, M.: Detecting recovery of the stratospheric ozone layer, Nature, 549, 211-218, https://doi.org/10.1038/nature23681, 2017.

Chipperfield, M. P., Dhomse, S., Hossaini, R., Feng, W., Santee, M. L., Weber, M., Burrows, J. P., Wild, J. D., Loyola, D., and Coldewey-Egbers, M.: On the cause of recent variations in lower stratospheric ozone, Geophys. Res. Lett., 45, 5718-5726, https://doi.org/10.1029/2018GL078071, 2018.

Duan, J., Tian, W., Zhang, J., Hu, Y., Yang, J., Wang, T., and Huang, R.: Impact of the Indian Ocean SST on wintertime total column ozone over the Tibetan Plateau, J. Geophys. Res., 128, e2022JD037850, https://doi.org/10.1029/2022JD037850, 2023.

Feng, W., Chipperfield, M. P., Roscoe, H. K., Remedios, J. J., Waterfall, A. M., Stiller, G. P., Glatthor, N., Höpfner, M., and Wang, D. Y.: Three-dimensional model study of the Antarctic ozone hole in 2002 and comparison with 2000, J. Atmos. Sci., 62, 822-837, https://doi.org/10.1175/JAS-3335.1, 2005.

Feng, W., Chipperfield, M. P., S. Dhomse, B. M. Monge-Sanz, X. Yang, K. Zhang, and M. Ramonet: Evaluation of cloud convection and tracer transport in a three-dimensional chemical transport model, Atmos. Chem. Phys., 11, 5783--5803, 2011.

Feng, W., Dhomse, S. S., Arosio, C., Weber, M., Burrows, J. P., Santee, M. L., and Chipperfield, M. P.: Arctic ozone depletion in 2019/20: Roles of chemistry, dynamics and the Montreal Protocol, Geophys. Res. Lett., 48, e2020GL091911, https://doi.org/10.1029/2020GL091911, 2021.

Griffin, D., Walker, K. A., Wohltmann, I., Dhomse, S. S., Rex, M., Chipperfield, M. P., Feng, W., Manney, G. L., Liu, J., and Tarasick, D.: Stratospheric ozone loss in the Arctic winters between 2005 and 2013 derived with ACE-FTS measurements, Atmos. Chem. Phys., 19, 577-601, https://doi.org/10.5194/acp-19-577-2019, 2019.

Kiladis, G. N., Diaz, H. F.: Global climatic anomalies associated with extremes in the Southern Oscillation. J. Clim., 2, 1069–1090, https://doi.org/10.1175/1520-0442(1989)002%3C1069:GCAAWE%3E2.0.CO;2, 1989

Kuttippurath, J., Kleinböhl, A., Bremer, H., Küllmann, H., Notholt, J., Sinnhuber, B.-M., Feng, W., and Chipperfield, M.: Aircraft measurements and model simulations of stratospheric ozone and N2O: implications for chemistry and transport processes in the models, J. Atmos. Chem., 66, 41-64, 10.1007/s10874-011-9191-4, 2010.

Li, J., Sun, C., and Jin, F. F.: NAO implicated as a predictor of Northern Hemisphere mean temperature multidecadal variability, Geophys. Res. Lett., 40, 5497-5502, https://doi.org/10.1002/2013GL057877, 2013.

Li, Y., Li, J., Zhang, W., Chen, Q., Feng, J., Zheng, F., et al.: Impacts of the Tropical Pacific Cold Tongue Mode on ENSO Diversity Under Global Warming. Journal of Geophysical Research: Oceans, 122, 8524–8542. https://doi.org/10.1002/2017JC013052, 2017.

Li, Y., Dhomse, S. S., Chipperfield, M. P., Feng, W., Chrysanthou, A., Xia, Y., and Guo, D.: Effects of reanalysis forcing fields on ozone trends and age of air from a chemical transport model, Atmos. Chem. Phys., 22, 10635-10656, https://doi.org/10.5194/acp-22-10635-2022, 2022.

Mishra, V., Smoliak, B. V., Lettenmaier, D. P., Wallace, J. M.: A prominent pattern of year-to-year variability in Indian summer monsoon rainfall. Proc. Natl Acad. Sci. USA 109, 7213–7217, 2012.

Pokharel, M., Guang, J., Liu, B., Kang, S., Ma, Y., Holben, B. N., Xia, X., Xin, J., Ram, K., and Rupakheti, D.: Aerosol properties over Tibetan Plateau from a decade of AERONET measurements: baseline, types, and influencing factors, J. Geophys. Res.-Atmos., 124, 13357–13374, https://doi.org/10.1029/2019JD031293, 2019.

Pyper, B. J. and Peterman, R. M.: Comparison of methods to account for autocorrelation in correlation analyses of fish data, Can. J. Fish. Aquat. Sci., 55, 2127-2140, https://doi.org/10.1139/f98-104, 1998.

Rösevall, J. D., Murtagh, D. P., Urban, J., Feng, W., Eriksson, P., and Brohede, S.: A study of ozone depletion in the 2004/2005 Arctic winter based on data from Odin/SMR and Aura/MLS, J. Geophys. Res., 113, https://doi.org/10.1029/2007JD009560, 2008.

Schubert, S. D. and Munteanu, M. J.: An analysis of tropopause pressure and total ozone correlations, Mon. Weather Rev., 116, 569-582, https://doi.org/10.1175/1520-0493(1988)116<0569:AAOTPA>2.0.CO;2, 1988.

Seidel, D. J., and Randel, W. J.: Variability and trends in the global tropopause estimated from radiosonde data, J. Geophys. Res., 111, D21101, https://doi.org/10.1029/2006JD007363, 2006.

Singleton, C. S., Randall, C. E., Chipperfield, M. P., Davies, S., Feng, W., Bevilacqua, R. M., Hoppel, K. W., Fromm, M. D., Manney, G. L., and Harvey, V. L.: 2002-2003 Arctic ozone loss deduced from POAM III satellite observations and the SLIMCAT chemical transport model, Atmos. Chem. Phys., 5, 597-609, https://doi.org/10.5194/acp-5-597-2005, 2005.

Swinbank, R., and A. O'Neill: A stratosphere–troposphere data assimilation system. Mon. Wea. Rev., 122, 686–702, 1994.

Sun, C., Li, J., Ding, R., and Jin, Z.: Cold season Africa–Asia multidecadal teleconnection pattern and its relation to the Atlantic multidecadal variability, Clim. Dyn., 48, 3903-3918, https://doi.org/10.1007/s00382-016-3309-y, 2017.

Tian, B., Yung, Y. L., Waliser, D. E., Tyranowski, T., Kuai, L., Fetzer, E. J., and Irion, F. W.: Intraseasonal variations of the tropical total ozone and their connection to the Madden-Julian Oscillation, Geophys. Res. Lett., 34, https://doi.org/10.1029/2007GL029451, 2007.

Thompson, D. W. J., Wallace, J. M: The Arctic oscillation signature in the wintertime geopotential height and temperature fields. Geophys. Res. Lett. 25, 1297–1300, 1998.

Thompson, D. W. J. and Wallace, J. M.: Regional Climate Impacts of the Northern Hemisphere Annular Mode, Science, 293, 85–89, 2001.

Thompson, D., Kennedy, J., Wallace, J. et al: A large discontinuity in the mid-twentieth century in observed global-mean surface temperature. Nature 453, 646–649, https://doi.org/10.1038/nature06982, 2008.

Trenberth, K. E.: A quasi-biennial standing wave in the southern hemisphere and interrelations with sea surface temperature. Q. J. R. Meteorol. Soc. 101, 55–74, 1975.

Trenberth, K. E., Zhang, Y., Fasullo, J. Y., Taguchi, S.: Climate variability and relationships between top-of-atmosphere radiation and temperatures on Earth. Journal of Geophysical Research: Atmospheres, 120, 3642–3659, 2015.

Varotsos, C., Cartalis, C., Vlamakis, A., Tzanis, C., and Keramitsoglou, I.: The long-term coupling between column ozone and tropopause properties, J. Clim., 17, 3843-3854, https://doi.org/10.1175/1520-0442(2004)017<3843:TLCBCO>2.0.CO;2, 2004.

von Storch, H. and Zwiers, F. W.: Statistical Analysis in Climate Research, Cambridge University Press, Cambridge, UK, 234–241, https://doi.org/10.1017/CBO9780511612336, 1999.

Wallace, J. M., Zhang, Y., and Bajuk, L.: Interpretation of interdecadal trends in northern hemisphere surface air temperature, J. Clim., 9, 249-259, https://doi.org/10.1175/1520-0442(1996)009<0249:IOITIN>2.0.CO;2, 1996.

Wang, B., Lee, JY., Kang, IS. et al: How accurately do coupled climate models predict the leading modes of Asian-Australian monsoon interannual variability?. Clim Dyn 30, 605–619, https://doi.org/10.1007/s00382-007-0310-5, 2008.

Woolnough, S. J., J. M. Slingo, and B. J. Hoskins: The Relationship between Convection and Sea Surface Temperature on Intraseasonal Timescales. J. Climate, 13, 2086–2104, https://doi.org/10.1175/1520-0442(2000)013<2086:TRBCAS>2.0.CO;2, 2000.

WMO: Climatic change Report of a working group of the Commission for Climatology. Technical note No. 79, Geneva, 66 pp., https://library.wmo.int/records/item/58659-climatic-change (last access: 7 November2023), 1966.

Zhang, D., McPhaden, M. J., and Lee, T.: Observed interannual variability of zonal currents in the equatorial Indian Ocean thermocline and their relation to Indian Ocean Dipole, Geophys. Res. Lett., 41, 7933–7941, 2014.

Zhao, Y., Duan, A., and Wu, G.: Interannual variability of late-spring circulation and diabatic heating over the Tibetan Plateau associated with Indian ocean forcing, Adv. Atmos. Sci., 35, 927-941, https://doi.org/10.1007/s00376-018-7217-4, 2018.

---

## Author Response (AR3)

**Response to the Referees #1-2's comments on "The impact of El Niño–Southern Oscillation on the total column ozone over the Tibetan Plateau" (egusphere-2023-1452)**

We thank Referees #1-2 for making detailed comments and very useful suggestions to improve the paper. The manuscript has been revised and further improved in response to the referee's comments and suggestions. Below is a point-by-point response (in black) to the comments (in blue) followed by the corresponding changes in the manuscript with track changes (*in italics*).

The line numbers for the changes correspond to the marked-up manuscript version.

Response to *Anonymous Referee #1*

*General comments*

This second revision of the paper successfully improved the quality of the paper, better clarifying several points that remained obscure after the first revision. However, there are some remaining issues that deserve attention. First of all, the use of correlation coefficients to explain causation, which is absolutely not enough. As I pointed out previously, the correlation coefficients actually indicate the presence of a similar time pattern in the two variables, but do not tell anything about the cause-effect relationship. Please have a look at Correlation vs Causation | Introduction to Statistics | JMP and the collection of spurious correlations from Tyler Vigen Spurious Correlations (tylervigen.com) Considering the methodology adopted here, it may be sufficient to point out this drawback and consider the correlation as a possible indication of a cause-effect relationship which needs to be demonstrated with other methods.

We thank again for the reviewer's comments and suggestions. We had improved our abstract and conclusion to infer causation. In addition to the correlation coefficient, we reiterate that we also used the composite method, hypsometric equation, and TOMCAT results to study the linkage between ENSO and TP TCO. We also had a caveat of correlation analysis in the first and second revised version [Lines 417–421].

To further verify our statistical results, we design and perform a series of model experiments to isolate the influence of ENSO. Since many studies has used Community Atmosphere Model version 4 (CAM4) developed by NCAR to isolate the SST–forced responses (e.g. Neale et al., 2013; Warner et al., 2020; Pausata et al., 2023), we will use CAM4 to support our findings. CAM4 is the atmospheric component of the Community Earth System Model (CESM). CAM4 has a horizontal resolution of 1.9° (latitude) × 2.5° (longitude) and 26 vertical levels. Three experiments are described in Table A1, all of which are same, but different SST forcings. The experiments were each run for 30 years, with the first 5 years excluded as a spin–up period. The remaining 25 years were used for the analysis.

As the tropopause height (TH) variability plays a key role in our potential mechanism, we display in Figure A1 the TH responses to El Niño and La Niña, averaged from December to May, when the response is significant (Figure 2 in manuscript). Figure A1 shows that El Niño (La Niña) generally correspond to a positive (negative) tropopause pressure differences over the almost the whole TP, which is in good agreement with the pattern of composite associated with El Niño (La Niña) events (Figure 6a–6b in manuscript). Note that the simulated intensity of tropopause pressure is stronger than that of composite in observations, which may be due to the model parameterization scheme over the TP is not perfect. Nevertheless, these simulations do support our statistical results.

Table A1 Description of experiments.

| Experiments | Description |
|---|---|
| R0 | Control run, using case F_2000. The solar forcing, carbon dioxide, ozone concentration, and aerosol are fixed at their level of 2000. Prescribed SST forcing used climatological present-day SST provided by NCAR. |
| R1 | El Niño sensitive run, same as R0, but with SST anomalies (Figure 7a) added in Pacific (30°S-30°N, 120°E-80°W) from December to May. |
| R2 | La Niña sensitive run, same as R0, but with SST anomalies (Figure 7b) added in Pacific (30°S-30°N, 120°E-80°W) from December to May. |

[Figure]

**Figure A1: Differences in tropopause height (hPa) (a) between the El Niño response of experiments R1–R0 and (b) La Niña response of experiments R2–R0. The differences over the dotted regions are statistical significance at the 90% confidence level. The black lines represent the boundary of the TP.**

An additional important note is that the authors analyse the differences between the mean of the events and the climate average, without considering: 1) whether the calculated differences are statistically significant; 2) the deviations around the means. The consideration of only the average values without inclusion of the variability around the mean could potentially mask the fact that the events and the climatological average could in reality belong to the same distribution; in addition, it is well known that the arithmetic mean is a parameter which is not resistant nor robust, which means that the presence of outliers could dramatically affect its value, and are not good if the population distribution is not normal. Considering these drawbacks and comments, my opinion is that the paper needs new major revisions before it can be accepted for publication.

The calculated differences are statistically significant, please see the following response (6[th] *Specific comment*).

The deviations have been considered, please see the following responses (7[th] – 8[th] *Specific comment*).

The composites associated with Figures 3–8 are all tested by the *t*–test, and its statistical significance indicate our results are significantly different from climatological average. Although there are some outliers, the scatterplot of Figure 9 shows most El Niño and La Niña events are coherent with composite results.

*Specific comments*

1. Lines 63-65: ENSO is a coupled climate phenomenon, involving changes not only in the ocean but also in the atmosphere which are not described here.

Revised. [Lines 57–59]: "*ENSO represents a periodic fluctuation in the tropical Pacific sea surface temperature (oceanic part: El Niño) and sea level pressure of the overlying atmosphere (atmospheric part: Southern Oscillation) during the warm (El Niño) and cold phase (La Niña)*".

2. Lines 112-115: This means that you are describing just the oceanic branch.

The Niño 3.4 index we used can measure ENSO, including its oceanic and atmospheric branches (e.g. Trenberth, 1997).

3. Lines 197-200 and previous review: the correlation coefficient is not indicative of a cause-effect relationship. Or, put in another way, "correlation is not causation". See: https://www.jmp.com/en_au/statistics-knowledge-portal/what-is-correlation/correlation-vs-causation.html So, I would suggest rephrasing or improving the analysis to infer causation.

Yes. In the second revision, the context had been improved to infer causation as shown in [Lines 194–196]: "……*response of TCO over the TP to ENSO may extend from winter of ENSO's mature phase to spring*……". In addition, we had added a caveat of correlation analysis in the second revision [Lines 417–421]. Please see more responses to *General comment*.

4. Figure 2: I think the Figure needs better discussion and presentation, as: 1) it is presented as lagged correlation, but where is the lag? 2) the correlation is never higher than 0.40 (not high). Also, I do not understand why the correlation decreases without QBO.

To make it clear, more responses are shown as follows:

1) In the top X-axis, positive leads indicate that ENSO is leading or TP TCO is lagging. We have added it into Figure 2 caption [Line 217].

2) After removing the ENSO signal, there is no significant TCO anomaly over the whole TP during El Niño and La Niña years for C3S dataset and TOMCAT simulation (Figure A1 in the second response). Comparing the composite analyses of TP TCO with and without ENSO signal, it is further suggested that ENSO has a potential influence on TP TCO although their correlation is not very high. This is similar to the correlation between ENSO and monsoon. For example, ENSO plays a vital role in the American monsoon system although their correlation is lower than 0.4 (e.g. Arias et al., 2015).

3) The correlation without QBO signal (Figure 2b) is stronger than raw correlation (Figure 2a), which may be related to exclude interference of QBO signal.

5. Figure 3: The spatial pattern and also the values of anomalies of C3S and TOMCAT are consistently different. This needs to be better discussed in the text.

Good point. We have added the discussion into the revision. [Lines 224–228]: "*Although the spatial pattern and also the values of anomalies of C3S and TOMCAT are consistently different (**Figure 3**) due to the model's biases (Dhomse et al., 2021), the El Niño (La Niña) events correspond to the significantly positive (negative) TCO anomalies over the whole TP in both the C3S dataset and TOMCAT results (**Figure 3**). This result is consistent with that of correlation coefficients (**Figure 2**)……*".

6. Lines 273-280 and Figure 5: Are the differences statistically significant? The plots do not show very large differences, and the reader might wonder whether they are statistically significant. Also, since composite means are plotted, it would be needed to associate standard deviations / errors to the mean to be consistent (the difference could fall within the deviation range from the climate mean).

Yes, the differences are statistically significant. Figure 5 is identical to Figure 4 (Fig 4 → composite percentage change of ozone profile; Fig 5 → composite of ozone profile). We had added statistically significant and standard deviation (SD) into Figure 4 in the first revision.

7. Figure 6: Same as above for Figure 5 as for the climate mean lines.

We have modified the Figure 6 in the new revision. Green lines with standard deviation (shaded areas) indicate the composite tropopause, and thick green lines indicate the composite tropopause which exceed the 90% confidence level.

8. Lines 354-356: Same as above as for the calculation of only means without considering a deviation around the mean value.

Based on Figure 8, Figure A2 shows the area-averaged values with standard deviation (vertical bars) of Tthickness and Ttroposphere_mean over the whole TP. We have added this Figure into Supplementary Material.

[Figure]

**Figure A2: The circles represent the area–averaged values of Tthickness and Ttroposphere_mean over the whole TP based on Figure 8. The vertical bars represent standard deviation.**

9. Figure 9: Again, only mean values are considered in the analysis.

Figure 9 is scatterplot, and it mainly focuses on the relationship between ENSO events and air thickness as well as tropopause rather than their mean values. Please see the context as shown in [Lines 356–371].

***Technical comments***

1. Lines 34-37: The two sentences are somehow a repetition, I think a rephrase could improve the readability.

Rephrased.

2. Line 64: change "warmer phase (El Niño) and colder phase (La Niña)" to " the warm (El Niño) and cold phase (La Niña)"

Changed.

3. Line 73: Add "a very limited number of" before "events"

Added

4. Lines 215-216: It is not clear which results show this minor impact of QBO. Please explain better, perhaps moving sentences.

Deleted.

5. Line 255 and 278: Which biases cause this disagreement? Explain.

We had explained the biases in [Lines 189–191].

6. Lines 264-265: It is not clear which are the shaded regions, as the colours refer to the color scale.

Changed. [Lines 244-245]: "*Coloured regions indicate statistical significance……*"

7. Line 273: Delete "the" after "Such".

Deleted.

8. Line 283: Change "of" to "in"

Changed.

9. Lines 411-415: This scheme with arrows could be brilliant for an oral presentation over a slide, but not so much for a paper. Please rephrase with complete sentences.

OK. It has been deleted.

**References**

Arias, P.A., Fu, R., Vera, C. et al.: A correlated shortening of the North and South American monsoon seasons in the past few decades. Clim. Dyn., 45, 3183–3203, https://doi.org/10.1007/s00382-015-2533-1, 2015.

Neale, R. B., Richter, J., Park, S., Lauritzen, P. H., Vavrus, S. J., Rasch, P. J., and Zhang, M.: The Mean Climate of the Community Atmosphere Model (CAM4) in Forced SST and Fully Coupled Experiments. Journal of Climate, 26(14), 5150-5168, https://doi.org/10.1175/JCLI-D-12-00236.1, 2013.

Pausata, F. S. R., Zhao, Y., Zanchettin, D., Caballero, R., and Battisti, D. S.: Revisiting the mechanisms of ENSO response to tropical volcanic eruptions. Geophysical Research Letters, 50, e2022GL102183. https://doi.org/10.1029/2022GL102183, 2023.

Trenberth, K. E.: The Definition of El Niño. Bull. Amer. Meteor. Soc., 78, 2771–2778, https://doi.org/10.1175/1520-0477(1997)078<2771:TDOENO>2.0.CO;2, 1997.

Warner, J. L., Screen, J. A., and Scaife, A. A.: Links between Barents-Kara sea ice and the extratropical atmospheric circulation explained by internal variability and tropical forcing. Geophysical Research Letters, 47, e2019GL085679. https://doi.org/10.1029/2019GL085679, 2020.

Response to *Anonymous Referee #2*

*General comments*

The manuscript of "The impact of El Niño–Southern Oscillation on the total column ozone over the Tibetan Plateau" by Yang Li et al. focuses on the effect of the ENSO on total column ozone (TCO) above the Tibetan Plateau (TP). Through analysis of the aggregated long-term satellite data C3S, the chemical transport model TOMCAT, and the water vapor and ozone dataset SWOOSH, the authors present statistical significant relations of positive TCO anomalies above the TP during El Niño and vice versa for La Niña events. These anomalies are robustly found in all presented datasets.
This study is definitely relevant and suits the scope of ACP well and is a welcome addition for the atmospheric community. The manuscript itself is mostly well written and understandable. However, there are parts, that are less well described and early conclusions that cannot be followed in the current state of the article. Hence, a major revision of this article are needed before a recommendation for publishing can be made.

Many thanks for the positive evaluation and the suggestions to improve the paper. We have revised our manuscript based on your comments and suggestion.

In particular, the authors only look at correlations of different parameters and jump to the conclusion of a causation. The whole mechanism of the TCO increase described in the conclusion is not well supported by the findings of the study and could be argued for in different order as well. The ENSO is leading the TCO anomaly but it is not clear if the cooling of the troposphere or the lowered geopotential height of the 150 hPa level is the next step in the mechanism. Since only the correlations are shown, this conclusion cannot be drawn.

We had improved our abstract and conclusion to infer causation. We also had added a caveat of correlation analysis in the second revision [Lines 417–421].

To verify this potential mechanism, we design and perform a series of model experiments to isolate the influence of ENSO. Since many studies has used Community Atmosphere Model version 4 (CAM4) developed by NCAR to isolate the SST–forced responses (e.g. Neale et al., 2013; Warner et al., 2020; Pausata et al., 2023), we will use CAM4 to support our findings. CAM4 is the atmospheric component of the Community Earth System Model (CESM). CAM4 has a horizontal resolution of 1.9° (latitude) × 2.5° (longitude) and 26 vertical levels. Three experiments are described in Table B1, all of which are same, but different SST forcings. The experiments were each run for 30 years, with the first 5 years excluded as a spin–up period. The remaining 25 years were used for the analysis.
As the cooling/warming troposphere is accompanied by the tropopause height (TH) variability and the TH plays a key role in our potential mechanism, we display in Figure B1 the TH responses to El Niño and La Niña, averaged from December to May, when the response is significant (Figure 2 in manuscript). Figure B1 shows that El Niño (La Niña) generally correspond to a positive

(negative) tropopause pressure differences over the almost the whole TP, which is in good agreement with the pattern of composite associated with El Niño (La Niña) events (Figure 6a–6b in manuscript). Note that the simulated intensity of tropopause pressure is stronger than that of composite in observations, which may be due to the model parameterization scheme over the TP is not perfect. Nevertheless, these simulations do support our statistical results.

Table B1 Description of experiments.

| Experiments | Description |
|---|---|
| R0 | Control run, using case F_2000. The solar forcing, carbon dioxide, ozone concentration, and aerosol are fixed at their level of 2000. Prescribed SST forcing used climatological present-day SST provided by NCAR. |
| R1 | El Niño sensitive run, same as R0, but with SST anomalies (Figure 7a) added in Pacific (30°S-30°N, 120°E-80°W) from December to May. |
| R2 | La Niña sensitive run, same as R0, but with SST anomalies (Figure 7b) added in Pacific (30°S-30°N, 120°E-80°W) from December to May. |

[Figure]

**Figure B1: Differences in tropopause height (hPa) (a) between the El Niño response of experiments R1–R0 and (b) La Niña response of experiments R2–R0. The differences over the dotted regions are statistical significance at the 90% confidence level. The black lines represent the boundary of the TP.**

Furthermore, the increase of the TCO is argued by the reduction in tropospheric height and a subsequent "change in the relative amounts of ozone–poor tropospheric and ozone–rich stratospheric air in the profile". While this is true for the relative amounts of air, it is not necessarily for the TCO. The TCO can only increase if there is an influx of stratospheric, ozone-rich air from other regions. This is not talked about although it is the center point of this study.
In general, the authors should either rephrase their claims or increase their evidence for the presented mechanism. In the current state, the stated causation is not evident.
Following are specific and technical comments with citations from the manuscript given in italics.

Based on previous studies (e.g. Salby and Callaghan 1993; Appenzeller et al., 2000; Tian et al., 2007, 2008; Zhang et al., 2014, 2015), the increasing TH corresponds to the upward motion, which is accompanied by mass convergence at lower troposphere; this would carry ozone-poor tropospheric air into the UTLS region and hence decrease the TCO. For the decreasing TH, it

favours the downward motion and convergence at lower stratosphere, which could bring ozone-rich stratospheric air into the UTLS region, thereby giving rise to TCO increase (e.g. Salby and Callaghan 1993; Appenzeller et al., 2000; Tian et al., 2007, 2008; Zhang et al., 2014, 2015).
We have rephrased context as shown in Section 3.2 [Lines 273–276]: "*Therefore, the increasing (decreasing) TH will tend to carry ozone-poor (ozone-rich) tropospheric (stratospheric) air into the UTLS region, and thus decrease (increase) the partial column ozone, which in turn contributes to the TCO decrease (increase)*".

*Specific comments*

1. l 57 ff: There is no mention about the atmospheric part of the ENSO, i.e., the SO part. Add a sentence describing this as well.

Revised. [Lines 57–59]: "*ENSO represents a periodic fluctuation in the tropical Pacific sea surface temperature (oceanic part: El Niño) and sea level pressure of the overlying atmosphere (atmospheric part: Southern Oscillation) during the warm (El Niño) and cold phase (La Niña)*".

2. l 99: how many vertical levels does your data span between 1000 and 0.01 hPa?

Added. [Lines 100–101]: "…… from 1000 to 0.01 hPa (137 vertical levels)".

3. l 122:which is calculated according to cross correlation function (Chatfield, 1982): probably an article missing (a or the?), but the cross correlation function should be described by a bit more detail. Or the sentence should be rephrased for better understandability.

Rephrased. [Lines 124–126]: "……*which is calculated according to the cross correlation function (e.g. Trenberth et al., 2002). The significant cross correlation between ENSO(t) and TCO(t+$\tau$) indicates that ENSO may influence the TCO when ENSO leads TCO at a lead of $\tau$*".

4. l 224: ±1.2% percentage change (i.e., anomaly divided by climate mean) of TP TCO You stated that your dataset has an accuracy of about 2%, which would render this change to be within the expected error if found in an individual case. Please comment on the this discrepancy within the text. Since you use composite events, the errors are canceling to some extent. Do you have a feeling to what extent?

From the perspective of TCO, Xie et al. (2014) focused on the impacts of ENSO on global ozone change using ERA-Interim data, and their percentage change over the latitude-longitude TP domain (their Figure 6a) is consistent with ours. Because of similar change between our results and previous study, we feel the errors will be cancelled to some extent using composite although

we are not sure what extent. To be clearer, we modified the percentage change to TCO anomaly as shown in [Lines 228]: "……*about ±3.5 DU of TP TCO anomaly* ……"

From the perspective of ozone profile, Figure 4 shows that the ENSO events correspond to about ±6% percentage change at 200-70 hPa, which is larger than the systematic error (below 2%). We have added this into [Lines 251]: "……*about ±6%......*".

5. l 225 ff: To further clarify the influence of ENSO on TP TCO, we use regression method to remove the ENSO signal from TP TCO and then perform composite TCO during El Niño and La Niña years. Without considering the ENSO signal on the TP TCO, both C3S data and TOMCAT simulation show that there is no significant TCO anomalies over the whole TP during El Niño and La Niña years (not shown). As it is, this section gives no new information since it basically reads as: if we remove the linear relation, we find no longer a linear relation. Please describe how you removed the ENSO signal. This should be a physically justified approach for the separation. If it is actually a regression separation, remove this part.

We used regression method to remove the ENSO signal, and we have removed this part.

6. Fig. 3: There is a shaded region mentioned for the stat. significance, but not shown. Change this to state something like: 'There is no region, where the anomaly is statistically significant beyond the 90% confidence level.' in the caption.

Sorry for this unclear. The caption has been revised as shown in [Lines 243–244]: "*Coloured regions indicate statistical significance at the 90% confidence level……*".

7. l 243 ff: although the altitudes of 100–50 hPa agree with your vertical shift hypothesis, the Ozone profiles between 200–100 hPa show a different slope for El Niño and La Niña. This might not only be related to a vertical shift but a compression/stretching of the profile for Niño and La Niña, respectively?

These sentences associated with Figure 4 depict the composite percentage change (%) of ozone profile rather than the raw ozone profile. Although the slopes of percentage change between 100–50 hPa and 200–100 hPa are different, Figure 5 shows the ozone profiles at these two levels all show the downward (upward) for El Niño (La Niña) events. Sorry for this unclear. We have revised the sentences as shown in [Lines 250–251]: "…...*increase (decrease) of ozone percentage change (about ±6%) at 200–70 hPa*……".

8. l 270 ff: Therefore, a decrease of tropopause height (TH) will tend to replace ozone–poor tropospheric air by ozone–rich stratospheric air in the UTLS region ... This would only be true, if there is horizontal flow of from surrounding areas towards the TP. In other words, a constant

Stratopause would be expected and/or an opposite TCO anomaly adjacent to the TP. The Ozone has to come from somewhere. Please add on this question for a more detailed description of this process. As it is right now, your point is not self-sufficient.

Please see the detailed response mentioned above (3rd *General comment*).

9. l 331 f: That is, the tropospheric temperature will warm (cool) when the rising (falling) GH150 causes the increased (decreased) air column thickness. This sounds too much like a cause and consequence where only correlation is seen. Please rephrase this sentence accordingly.

Revised. [Lines 337-339]: "*That is, the increased (decreased) air column thickness associated with the rising (falling) GH150 favours the warming (cooling) tropospheric temperature.*"

10. l 339 f: According to equation (3), this implies that the El Niño (La Niña) events favour the decreased (increased) air thickness and thus contributes to the cooling (warming) tropospheric mean temperature (Figures 8). 'Contributes' is more than you can claim here, I would say it is 'associated with' since you only found the correlation.

Revised. [Lines 347]: "……*thus associated with*……".

11. l 351 f: ... there is a strong negative correlation (–0.56) between Niño 3.4 and temperature associated with air thickness. While it is very convincing that the air thickness is related to whether there is a El Niño or La Niña, the correlation to Niño 3.4 is not as convincing. Did you have a look at other indices and their correlation to the air thickness? And what about the non-ENSO events in the same periods, do they fall close to the center point in this diagram, or are they scattered on the vertical axis? Please comment on this in the text.

Good points! To make it clear, more responses are shown as follows:

1) In addition to Niño 3.4 index, Niño 3 index, which is calculated with SSTA in the area (5°S - 5°N, 90°W - 150°W), is another indicator for ENSO. The correlation coefficient between Niño 3 and temperature associated with air thickness is about –0.559 (p < 0.01), which is in good agreement with that of Niño 3.4 index. We have added this in [Lines 360-361]: "*Another ENSO index (Niño 3; area–averaged SSTA in 5°S–5°N, 90°W–150°W) has about the same correlation as Nino 3.4 index*".

2) We have added this in [Lines 361-363]: "*Meanwhile, the correlation coefficient between non– ENSO events (Niño 3.4 index is less than 0.5 K and greater than –0.5 K) and their corresponding temperature associated with air thickness is about –0.12 (p > 0.1), suggesting that there is no relationship between them*".

12. l 378 f: ...with and without ENSO signal... should be '...with and without QBO signal...' if I'm not mistaken.

Sorry for this typo. Yes, it is "*with and without QBO signal*", and it has been revised in [Lines 388–389].

13. l 393 ff: Thirdly, such a TH decrease tends to cause a change in the relative amounts of ozone–poor tropospheric and ozone–rich stratospheric air in the profile, which increases the partial column ozone in the UTLS... See above. The partial column ozone will change for a fixed pressure interval, but for the TCO, there needs to be an influx (or source) of Ozone adjacent to the column. The process described here is not clear. The main question I'm left with is: where is the additional Ozone in the TCO stemming from?

Please see the detailed response mentioned above (3[rd] *General comment*).

*Technical comments*

1. l 67: since the satellite era → during the satellite era

Revised.

2. l 104: of area averaged of SST anomalies (SSTA) → of area averaged SST anomalies (SSTA)

Revised.

3. l 104 f: consider adding a reference to the website here

We had added website in *Data availability* [Lines 429–430].

4. l 106 f: As the SWOOSH spans from 1984 to the present, its anomalies are with respect to the period 1984–2021. consider rephrasing this for better understanding

Rephrased.

5. l 131 f: In view of the fact that the relationship between the positive and negative ENSO phases may not be linear (An and Jin 2004), we consider El Niño and La Niña events should be analyzed separately. remove 'should be analyzed' or 'we consider'

Yes. We have removed '*should be analyzed*'.

6. l 187: presentation → representation

Changed.

7. sec 3.1: use lead-lagged (or lagged-lead if you prefer this) consistently in your text and captions

OK. We have modified text and caption.

8. l 247: remove Such

Modified.

9. l 250: remove , whose results are consistent with Figures 5a and 5d. Since its only a zoom it is not only consistent but identical. No need to clarify after stating it's a zoom.

Removed.

**References**

Appenzeller, C., Weiss, A. K., and Staehelin, J.: North Atlantic Oscillation modulates total ozone winter trends, Geophys. Res. Lett., 27, 1131–1134, https://doi.org/10.1029/1999GL010854, 2000.

Neale, R. B., Richter, J., Park, S., Lauritzen, P. H., Vavrus, S. J., Rasch, P. J., and Zhang, M.: The Mean Climate of the Community Atmosphere Model (CAM4) in Forced SST and Fully Coupled Experiments. Journal of Climate, 26(14), 5150-5168, https://doi.org/10.1175/JCLI-D-12-00236.1, 2013.

Pausata, F. S. R., Zhao, Y., Zanchettin, D., Caballero, R., and Battisti, D. S.: Revisiting the mechanisms of ENSO response to tropical volcanic eruptions. Geophysical Research Letters, 50, e2022GL102183. https://doi.org/10.1029/2022GL102183, 2023.

Salby, M. L. and Callaghan, P. F.: Fluctuations of total ozone and their relationship to stratospheric air motions, J. Geophys. Res., 98, 2715-2727, https://doi.org/10.1029/92JD01814, 1993.

Tian, B., Yung, Y. L., Waliser, D. E., Tyranowski, T., Kuai, L., Fetzer, E. J., and Irion, F. W.: Intraseasonal variations of the tropical total ozone and their connection to the Madden-Julian Oscillation, Geophys. Res. Lett., 34, https://doi.org/10.1029/2007GL029451, 2007.

Tian, W., Chipperfield, M., and Huang, Q.: Effects of the Tibetan Plateau on total column ozone distribution, Tellus B, 60, 622-635, https://doi.org/10.1111/j.1600-0889.2008.00338.x, 2008.

Warner, J. L., Screen, J. A., and Scaife, A. A.: Links between Barents-Kara sea ice and the extratropical atmospheric circulation explained by internal variability and tropical forcing. Geophysical Research Letters, 47, e2019GL085679. https://doi.org/10.1029/2019GL085679, 2020.

Xie, F., Li, J., Tian, W. et al.: The impacts of two types of El Niño on global ozone variations in the last three decades. Adv. Atmos. Sci. 31, 1113–1126, https://doi.org/10.1007/s00376-013-3166-0, 2014.

Zhang, J., Tian, W., Xie, F., Tian, H., Luo, J., Zhang, J., Liu, W., and Dhomse, S.: Climate warming and decreasing total column ozone over the Tibetan Plateau during winter and spring, Tellus B, 66, 23415, https://doi.org/10.3402/tellusb.v66.23415, 2014.

Zhang, J., Tian, W., Wang, Z., Xie, F., and Wang, F.: The influence of ENSO on northern midlatitude ozone during the winter to spring transition, J. Clim., 28, 4774-4793, https://doi.org/10.1175/JCLI-D-14-00615.1, 2015.

---

## Author Response (AR4)

**Response to the Referees #2-3's comments on "The impact of El Niño–Southern Oscillation on the total column ozone over the Tibetan Plateau" (egusphere-2023-1452)**

We thank Referees #2-3 for making detailed comments and very useful suggestions to improve the paper. The manuscript has been revised and further improved in response to the referee's comments and suggestions. Below is a point-by-point response (in black) to the comments (in blue) followed by the corresponding changes in the manuscript with track changes (*in italics*).

The line numbers for the changes correspond to the revision.

Response to *Anonymous Referee #2*

*General comments*

The article "The impact of El Niño–Southern Oscillation on the total column ozone over the Tibetan Plateau" by Yang Li et al. investigates anomalies of total column ozone over the Tibetan Plateau During El Niño and La Niña events. As for the newest revision, I recommend this article for publication after some minor changes (mostly technical).
Most issues have been resolved by the changes and the author's comments. In particular, the revised version has clarified the methodology and the text throughout the article. Furthermore, the revised text reflects the data and derivable results much better than before.
Following are some minor comments that should be addressed before acceptance to ACP.

We thank again for the reviewer's comments and suggestions.

*Minor comments*

1. Consider adding the discussion of Table A1/B1 and Figure A1/B1 to the Appendix (or at least to the supplement) since it supports your analysis and findings.

Good point. We have added the discussion into *Summary and discussion* of revision [Lines 397–409].

2. l 198: "QBO signal ($TCO_{removeQBO}$) can be written as follows:" is there a reference for the linear separation of the QBO effect? If so, please add it here.

Yes, we have added the reference (e.g. Randel et al., 2009) into [Lines 198–199].

3. Fig. 6 caption: The difference between "thick green lines" and "green lines" is barely visible. I suggest increasing the width of the thick line. Also consider shrinking the vertical axis in order to enhance visibility of the "shaded area" around the green lines.

Yes. We have increased the width and reduced the vertical axis. Please see the Figure 6 in revision.

*Technical comments*

1. l 195: ".. another potential important source of interannual variability." -¿ "another potential important source of interannual variability of TCO above the TP."

Rephrased.

2. l 335 "...thickness and thus associated with...": There is a word missing. E.g., either "are thus associated" or "thus are associated".

Added.

3. l 379 "...to MAM in decaying phase...": Either "in its decaying phase" or "in the decaying phase"

Rephrased.

**References**

Randel, W. J., Garcia, R. R., Calvo, N., and Marsh, D.: ENSO influence on zonal mean temperature and ozone in the tropical lower stratosphere, Geophys. Res. Lett., 36, https://doi.org/10.1029/2009GL039343, 2009.